# Free Lunch in Pathology Foundation Model: Task-specific Model Adaptation with Concept-Guided Feature Enhancement

**Yanyan Huang**
The University of Hong Kong
`yanyanh@connect.hku.hk`

**Weiqin Zhao**
The University of Hong Kong
`wqzhao98@connect.hku.hk`

**Yihang Chen**
The University of Hong Kong
`yihangc@connect.hku.hk`

**Yu Fu**
Lanzhou University
`fuyu@lzu.edu.cn`

**Lequan Yu**[*]
The University of Hong Kong
`lqyu@hku.hk`

## Abstract

Whole slide image (WSI) analysis is gaining prominence within the medical imaging field. Recent advances in pathology foundation models have shown the potential to extract powerful feature representations from WSIs for downstream tasks. However, these foundation models are usually designed for general-purpose pathology image analysis and may not be optimal for specific downstream tasks or cancer types. In this work, we present ***Concept Anchor-guided Task-specific Feature Enhancement*** (CATE), an adaptable paradigm that can boost the expressivity and discriminativeness of pathology foundation models for specific downstream tasks. Based on a set of task-specific concepts derived from the pathology vision-language model with expert-designed prompts, we introduce two interconnected modules to dynamically calibrate the generic image features extracted by foundation models for certain tasks or cancer types. Specifically, we design a Concept-guided Information Bottleneck module to enhance task-relevant characteristics by maximizing the mutual information between image features and concept anchors while suppressing superfluous information. Moreover, a Concept-Feature Interference module is proposed to utilize the similarity between calibrated features and concept anchors to further generate discriminative task-specific features. The extensive experiments on public WSI datasets demonstrate that CATE significantly enhances the performance and generalizability of MIL models. Additionally, heatmap and umap visualization results also reveal the effectiveness and interpretability of CATE. The source code is available at https://github.com/HKU-MedAI/CATE.

## 1 Introduction

Multiple Instance Learning (MIL) [26, 34, 23, 2] is widely adopted for weakly supervised analysis in computational pathology, where the input of MIL is typically a set of patch features generated by a pre-trained feature extractor (*i.e.*, image encoder). Although promising progress has been achieved, the effectiveness of MIL models heavily relies on the quality of the extracted features. A robust feature extractor can discern more distinctive pathological features, thereby improving the predictive capabilities of MIL models. Recently, several studies have explored using pretrained foundation models on large-scale pathology datasets with self-supervised learning as the feature extractors for WSI analysis [37, 7, 3, 36]. Additionally, drawing inspiration from the success of Contrastive

---

[*]Corresponding Author.

38th Conference on Neural Information Processing Systems (NeurIPS 2024).

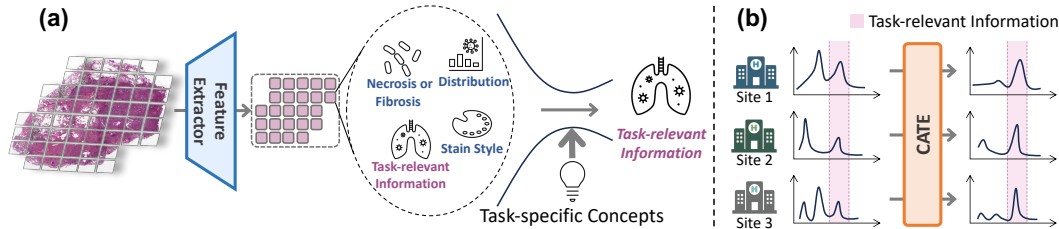

Figure 1: (a) Illustration of the key idea of concept-guided information bottleneck to enhance the task-relevant information and discard the task-irrelevant information. (b) Task-specific model adaptation with CATE to enhance the generalization across different data sources.

Language-Image Pretraining (CLIP) [30, 20] in bridging visual and linguistic modalities, some works have aimed to develop a pathology vision-language foundation model (VLM) to simultaneously learn representations of pathology images and their corresponding captions [15, 25]. The intrinsic consistency between the image feature space and caption embedding space in the pathology VLM enables the image encoder to extract more meaningful and discriminative features for downstream WSI analysis applications [25].

Although the development of these pathology foundation models has significantly advanced computational pathology, these models are designed for general-purpose pathology image analysis and may not be optimal for specific downstream tasks or cancer types, as the features extracted by the image encoder may contain generic yet task-irrelevant information that will harm the performance of specific downstream tasks. For example, as illustrated in Figure 1(a), the features extracted by the image encoder of a pathology VLM can include both task-relevant information (*e.g.*, arrangement or morphology of tumor cells) and task-irrelevant elements(such as background information, stain styles, etc.). The latter information may act as "noise", distracting the learning process of MIL models tailored to specific tasks, and potentially impairing the generalization performance of these models across different data sources. Consequently, it is crucial to undertake task-specific adaptation to enhance feature extraction of generic foundation models and enable MIL models to concentrate on task-relevant information and thus improve analysis performance and generalization [32, 38].

In this paper, we propose a novel paradigm, named **C**oncept **A**nchor-guided **T**ask-specific Feature **E**nhancement (CATE), to enhance the generic features extracted by the pathology VLM for specific downstream tasks (*e.g.*, cancer subtyping). Without requiring additional supervision or significant computational resources, CATE offers an approximately *"free lunch"* in the context of pathology VLM. Specifically, we first derive a set of task-specific concept anchors from the pathology VLM with task-specific prompts, and these prompts rely on human expert design or are generated through querying large language models (LLMs), necessitating a certain level of pathological background knowledge. Based on these concept anchors, we design two concept-driven modules, *i.e.*, the Concept-guided Information Bottleneck (CIB) module and the Concept-Feature Interference (CFI) module, to calibrate and generate task-specific features for downstream analysis. Particularly, with the task-specific concepts as the guidance, the CIB module enhances task-relevant features by maximizing the mutual information between the image features and the concept anchors and also eliminates task-irrelevant information by minimizing the superfluous information, as shown in Figure 1(a). Moreover, the CFI module further generates discriminative task-specific features by utilizing the similarities between the calibrated image features and concept anchors (*i.e.*, concept scores). By incorporating the CATE into existing MIL frameworks, we not only obtain more discriminative features but also improve generalization regarding domain shift by eliminating task-irrelevant features and concentrating on pertinent information, as shown in Figure 1(b).

In summary, the main contributions of this work are threefold:

- We introduce a novel method, named **CATE**, for model adaptation in computational pathology. To the best of our knowledge, this is the first initiative to conduct *task-specific* feature enhancement based on the pathology foundation model for MIL tasks.

- We design a new **CIB** module to enhance the task-relevant information and discard irrelevant information with the guidance of task-specific concepts, and a new **CFI** module to generate task-specific features by exploiting the similarities between image features and concept anchors.

- Extensive experiments on Whole Slide Image (WSI) analysis tasks demonstrate that CATE significantly enhances the performance and generalization capabilities of MIL models.

## 2    Related Work

**Multiple Instance Learning (MIL) for WSI Analysis.** MIL is the predominant paradigm for WSI analysis, treating each WSI as a bag of patch instances and classifying the entire WSI based on aggregated patch-level features. Attention-based methods [16, 26, 17, 39] are highly regarded for their ability to determine the significance of each instance within the bag. For instance, Ilse *et al.* [16] introduced an attention-based MIL model, while Lu *et al.* [26] proposed clustering-constrained-attention to refine this mechanism further. To model the relationships among instances, graph-based and Transformer-based methods have been developed [10, 2, 13]. For example, Chen *et al.* [2] introduced a Transformer-based hierarchical network to capitalize on the inherent hierarchical structure of WSIs.

**Pathology Foundation Model.** With the advancement of foundation models in computer vision, several pathology foundation models have been developed to serve as robust image encoders for WSI analysis. Riasatian *et al.* [31] proposed fine-tuning the DenseNet [12] on the TCGA dataset, while Filiot *et al.* [7] utilized iBOT [42] to pretrain a vision Transformer using the Masked Image Modeling framework. Recently, Chen *et al.* [3] pre-trained a general-purpose foundation model on large-scale pathology datasets using DINOv2 [28], which has demonstrated strong and readily usable representations for WSI analysis. Inspired by CLIP [30], Ikezogwo *et al.* [15], Huang *et al.* [14], and Lu *et al.* [25] developed vision-language foundation models by training on large-scale pathology datasets with image-caption pairs. These foundation models have demonstrated superior performance in downstream tasks due to their ability to extract more discriminative features for WSI analysis.

**Feature Enhancement in Computational Pathology.** Several methods have been developed to obtain more discriminative features for WSI analysis by adapting pathology foundation models [41, 24] or designing new plug-and-play modules [35]. For instance, Zhang *et al.* [41] suggested aligning the image features with text features extracted from a pre-trained natural language model to enhance the feature representation of WSI patch images, while it operates solely at the patch level, without considering the informational relationship between image and text features. Recently, Tang *et al.* [35] introduced Re-embedded Regional Transformer for feature re-embedding, aimed at enhancing WSI analysis when integrated with existing MIL methods. However, while this method considers the spatial information of WSIs and adds flexibility to MIL models, it falls short in extracting task-specific discriminative information for WSI analysis.

## 3    Method

### 3.1    Overview

The proposed CATE can be seamlessly integrated with any MIL framework to adapt the existing pathology foundation model (Pathology VLM) for performance-improved WSI analysis via task-specific enhancement. Specifically, consider a training set $\mathcal{D} = \{(\mathbf{x}, \mathbf{y})\}$ of WSI-label pairs, where $\mathbf{x} = \{\boldsymbol{x}_1, \boldsymbol{x}_2, ..., \boldsymbol{x}_N\}$ is a set of patch features with dimension of $C$ (*i.e.*, $\boldsymbol{x}_i \in \mathbb{R}^C$) extracted by the image encoder of pathology VLM, $N$ denotes the number of patches, and $\mathbf{y}$ is the corresponding label. The objective of CATE is to obtain the corresponding enhanced task-specific feature set $\mathbf{z}$ from the original feature $\mathbf{x}$ with the guidance of pre-extracted concepts anchors $\mathbf{c}$ (see description below) for downstream usage:

$$\mathbf{z} = \textbf{CATE}\left(\mathbf{x}, \mathbf{c}\right), \hat{\mathbf{y}} = \text{MIL}\left(\mathbf{z}\right). \tag{1}$$

As illustrated in Figure 2, we design two different modules to enhance the extracted features from foundation models: (1) Concept-guided Information Bottleneck (CIB) module calibrates original image features with the guidance of concept anchors with information bottleneck principle; and (2) Concept-Feature Interference (CFI) module generates discriminative task-specific features by leveraging the similarities between the calibrated image features and concept anchors. Specifically, the enhanced patch features can be represented as $\mathbf{z} = \{\boldsymbol{z}_1, \boldsymbol{z}_2, ..., \boldsymbol{z}_N\}$, where $\boldsymbol{z}_i$ is the concatenation of the calibrated feature $\boldsymbol{\alpha}_i$ and the interference feature $\boldsymbol{\beta}_i$ generated by CIB and CFI module:

$$\boldsymbol{z}_i = \text{Concat}\left[\boldsymbol{\alpha}_i, \boldsymbol{\beta}_i\right] = \text{Concat}\left[\textbf{CIB}\left(\boldsymbol{x}_i, \mathbf{c}\right), \textbf{CFI}\left(\boldsymbol{x}_i, \mathbf{c}\right)\right]. \tag{2}$$

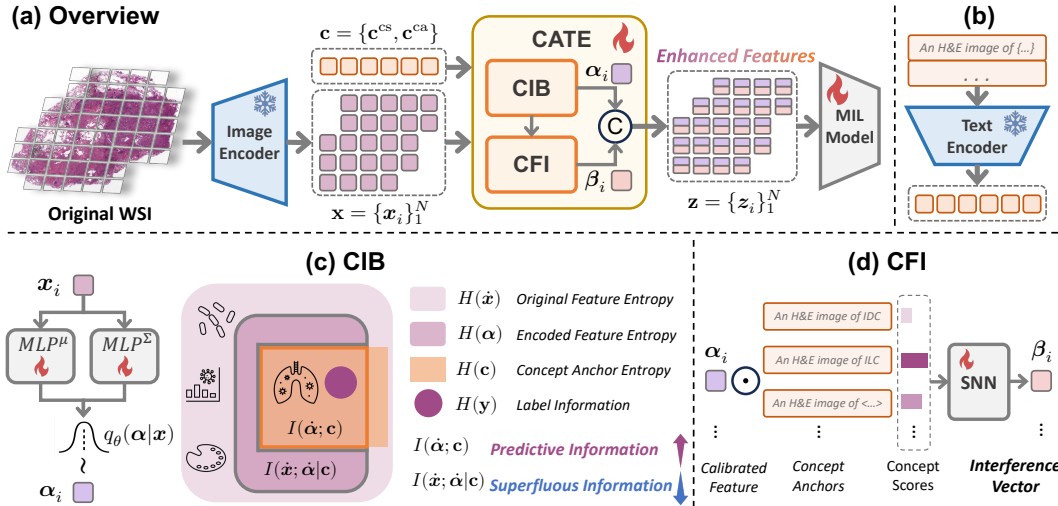

Figure 2: (a) Overview of CATE: the outputs of the CIB and CFI modules are concatenated to form the enhanced feature for downstream MIL models. (b) Task-relevant concept generation. (c) Concept-guided Information Bottleneck (CIB) module. (c) Concept-Feature Interference (CFI) module.

**Concept Extraction.** We extract two kinds of task-specific concept anchors, $\mathbf{c} = \{\mathbf{c}^{\text{cs}}, \mathbf{c}^{\text{ca}}\}$, comprising class-specific concepts $\mathbf{c}^{\text{cs}} = \{c_i^{\text{cs}}\}_{i=1}^m$ (*e.g.*, subtyping classes) and class-agnostic concepts $\mathbf{c}^{\text{ca}} = \{c_i^{\text{ca}}\}_{i=1}^n$ (*e.g.*, adipose, connective, and normal tissues), with $m$ and $n$ representing the numbers of class-specific and class-agnostic concepts, respectively. These concepts are generated by the text encoder of pathology VLM with prompt $\mathbf{p}$. Each prompt consists of a class name (*e.g.*, "invasive ductal carcinoma") and a template (*e.g.*, "An image of <*CLASSNAME*>"). To obtain more robust concepts, we use multiple prompts for each class and the final concept anchor is the average of the embeddings generated by different prompts. Details of class names and templates for various tasks are provided in Appendix G. Note that due to the inherent consistency between the image and text embedding space in VLM, these extracted concepts can also be regarded as image concept vectors.

## 3.2 Concept-guided Information Bottleneck

The objective of this module is to find a distribution $p(\boldsymbol{\alpha}|\mathbf{x})$ that maps the original image feature $\mathbf{x}$ into a representation $\boldsymbol{\alpha}$, which contains enhanced task-discriminative characteristics and suppressed task-irrelevant information. WSIs typically contain various cell types or tissues (*e.g.*, tumor cells, normal cells, adipose tissue, connective tissue), while only a subset of patches (*e.g.*, with tumor cells) is crucial for certain tasks such as tumor subtyping. We thus define $\hat{\mathbf{x}} = \{\hat{\boldsymbol{x}}_i\}_{i=1}^k \subseteq \mathbf{x}$ as the representative subset of the original feature set (*e.g.*, tumor tissue patches), where $k$ denotes the number of representative patches. Note that this selection can be conducted with a simple comparison of image features with class-specific concepts (see discussion below). To this end, the corresponding enhanced feature set is $\hat{\boldsymbol{\alpha}} = \{\hat{\boldsymbol{\alpha}}_i\}_{i=1}^k \subseteq \boldsymbol{\alpha}$ and we want to find the conditional distribution $p(\hat{\boldsymbol{\alpha}}_i|\hat{\boldsymbol{x}}_i)$ to map the selected patch feature $\hat{\boldsymbol{x}}_i \in \mathbb{R}^C$ into a more discriminative enhanced feature $\hat{\boldsymbol{\alpha}}_i \in \mathbb{R}^C$, which is discriminative enough to identify the label $\mathbf{y}$.

**Sufficiency and Consistency Requirements.** To quantify the informativeness requirement of the calibrated feature $\hat{\boldsymbol{\alpha}}$, we consider the **sufficiency** of $\hat{\boldsymbol{\alpha}}$ for $\mathbf{y}$. As defined in Appendix E.1, the encoded feature $\hat{\boldsymbol{\alpha}}$ derived from the original feature $\hat{\boldsymbol{x}}$ is sufficient for determining the label $\mathbf{y}$ if and only if the amount of task-specific information remains unchanged after calibration, *i.e.*, $I(\hat{\boldsymbol{x}}; \mathbf{y}) = I(\hat{\boldsymbol{\alpha}}; \mathbf{y})$. However, the label $\mathbf{y}$ pertains to the slide level and specific labels cannot be assigned to each instance due to the absence of patch-level annotations.

To address this challenge, we propose using the task-specific concept anchor as the guidance for each single $\hat{\boldsymbol{\alpha}}$. Specifically, we posit that the concept anchor $\mathbf{c}$ is distinguishable for the task and contains task-relevant information for label $\mathbf{y}$. Given the consistency between image and text features in pathology VLM, any representation $\hat{\boldsymbol{\alpha}}$ containing all information accessible from both image feature $\hat{\boldsymbol{x}}$ and concept $\mathbf{c}$ will also encapsulate the discriminative information required for the

label. This **consistency** requirement is detailed in Appendix E.1. Thus, if $\hat{\alpha}$ is sufficient for $\mathbf{c}$ (*i.e.*, $I(\hat{\boldsymbol{x}}; \mathbf{c}|\hat{\alpha}) = 0$), then $\hat{\alpha}$ is as predictive for label $\mathbf{y}$ as the joint of original feature $\hat{\boldsymbol{x}}$ and concept anchor $\mathbf{c}$. Applying the chain rule of mutual information, we derive:

$$I(\hat{\boldsymbol{x}}; \hat{\boldsymbol{\alpha}}) = \underbrace{I(\hat{\boldsymbol{\alpha}}; \mathbf{c})}_{\textcolor{magenta}{\textbf{Predictive Information}}} + \underbrace{I(\hat{\boldsymbol{x}}; \hat{\boldsymbol{\alpha}}|\mathbf{c})}_{\textcolor{blue}{\textbf{Superfluous Information}}} . \tag{3}$$

According to the consistency between the concept anchor and original feature, the mutual information term $I(\hat{\boldsymbol{\alpha}}; \mathbf{c})$ represents the predictive information for the task, while the conditional information term $I(\hat{\boldsymbol{x}}; \hat{\boldsymbol{\alpha}}|\mathbf{c})$ denotes task-irrelevant information (*i.e.*, superfluous information) in original patch feature $\hat{\boldsymbol{x}}$, which can be minimized to enhance the robustness and generalization ability of downstream MIL models. As a result, the main objective of the feature calibration in this module can be formalized as *maximize predictive information* $I(\hat{\boldsymbol{\alpha}}; \mathbf{c})$ while *minimize the superfluous information* $I(\hat{\boldsymbol{x}}; \hat{\boldsymbol{\alpha}}|\mathbf{c})$.

**Predictive Information Maximization (PIM).** The predictive information in Equ (3) equals to the mutual information between the calibrated feature and concept anchors. To maximize this, we choose the InfoNCE [27] to estimate the lower bound of the mutual information, which can be obtained by comparing positive pairs sampled from the joint distribution $\hat{\boldsymbol{\alpha}}, \boldsymbol{c}_{\text{pos}}^{\text{cs}} \sim p(\hat{\boldsymbol{\alpha}}, \boldsymbol{c}^{\text{cs}})$ to pairs $\hat{\boldsymbol{\alpha}}, \boldsymbol{c}_{j}^{\text{cs}}$ and $\hat{\boldsymbol{\alpha}}, \boldsymbol{c}_{j}^{\text{ca}}$ built using a set of negative class concepts $\boldsymbol{c}_{j}^{\text{cs}} \sim p(\boldsymbol{c}^{\text{cs}})$ and class-agnostic concepts $\boldsymbol{c}_{j}^{\text{ca}} \sim p(\boldsymbol{c}^{\text{ca}})$:

$$I_{NCE}(\hat{\boldsymbol{\alpha}}; \mathbf{c}) = \mathbb{E}_{\hat{\boldsymbol{\alpha}}, \boldsymbol{c}_{\text{pos}}^{\text{cs}}} \left[ \sum_{i=1}^{k} \log \frac{f\left(\boldsymbol{c}_{\text{pos}}^{\text{cs}}, \hat{\boldsymbol{\alpha}}_i\right)}{\sum_{j=1}^{m} f\left(\boldsymbol{c}_{j}^{\text{cs}}, \hat{\boldsymbol{\alpha}}_i\right) + \sum_{j=1}^{n} f\left(\boldsymbol{c}_{j}^{\text{ca}}, \hat{\boldsymbol{\alpha}}_i\right)} \right]. \tag{4}$$

We set $f(\boldsymbol{c}, \hat{\boldsymbol{\alpha}}_i) = \exp\left(\hat{\boldsymbol{\alpha}}_i^{\top} \boldsymbol{c}/\tau\right)$ with $\tau > 0$ in practice following [27]. By maximizing this mutual information lower bound, $f(\boldsymbol{c}, \hat{\boldsymbol{\alpha}}_i)$ will be proportional to the density ratio $p(\boldsymbol{c}, \hat{\boldsymbol{\alpha}}_i)/p(\boldsymbol{c}) p(\hat{\boldsymbol{\alpha}}_i)$ as proved in [27]. Hence, $f(\boldsymbol{c}, \hat{\boldsymbol{\alpha}}_i)$ preserves the mutual information between the calibrated feature and concept anchor. The detailed derivation can be found in Appendix E.2. The loss function for PIM can be denoted as:

$$\mathcal{L}_{PIM} = \mathbb{E}_{\hat{\boldsymbol{\alpha}}, \boldsymbol{c}_{\text{pos}}^{\text{cs}}} \left[ -\sum_{i=1}^{k} \frac{\hat{\boldsymbol{\alpha}}_i^{\top} \boldsymbol{c}_{\text{pos}}^{\text{cs}}}{\tau} \right] + \mathbb{E}_{\hat{\boldsymbol{\alpha}}, \boldsymbol{c}_{\text{pos}}^{\text{cs}}} \left[ \sum_{i=1}^{k} \log \left( \sum_{j=1}^{m} \exp \frac{\hat{\boldsymbol{\alpha}}_i^{\top} \boldsymbol{c}_{j}^{\text{cs}}}{\tau} + \sum_{j=1}^{n} \exp \frac{\hat{\boldsymbol{\alpha}}_i^{\top} \boldsymbol{c}_{j}^{\text{ca}}}{\tau} \right) \right]. \tag{5}$$

**Superfluous Information Minimization (SIM).** To compress task-irrelevant information, we aim to minimize the superfluous information term as defined in Equ (3). This objective can be achieved by minimizing the mutual information $I(\hat{\boldsymbol{x}}; \hat{\boldsymbol{\alpha}})$. In practice, we conduct SIM for *all patches* in the subset $\mathbf{x}$, as each patch may contain task-irrelevant information. Following [1], it can be represented as:

$$I(\boldsymbol{x}; \boldsymbol{\alpha}) = \int p(\boldsymbol{x}, \boldsymbol{\alpha}) \log p(\boldsymbol{\alpha}|\boldsymbol{x}) \, d\boldsymbol{x} d\boldsymbol{\alpha} - \int p(\boldsymbol{\alpha}) \log p(\boldsymbol{\alpha}) \, d\boldsymbol{\alpha}. \tag{6}$$

After that, we let the distribution of $\boldsymbol{\alpha}$: $r(\boldsymbol{\alpha})$ (*e.g.*, Gaussian distribution in this work), be a variational approximation to the marginal distribution $p(\boldsymbol{\alpha})$, and we can obtain the upper bound for $I(\boldsymbol{x}; \boldsymbol{\alpha})$:

$$I(\boldsymbol{x}; \boldsymbol{\alpha}) \leq \int p(\boldsymbol{x}) p(\boldsymbol{\alpha}|\boldsymbol{x}) \log \frac{p(\boldsymbol{\alpha}|\boldsymbol{x})}{r(\boldsymbol{\alpha})} d\boldsymbol{x} d\boldsymbol{\alpha}. \tag{7}$$

Furthermore, we use a variational distribution $q_\theta(\boldsymbol{\alpha}|\boldsymbol{x})$ with parameter $\theta$ to approximate $p(\boldsymbol{\alpha}|\boldsymbol{x})$ and we implement the parameterization of the variational distribution with MLP by predicting the mean and variance of the Gaussian distribution and sample the calibrated feature $\boldsymbol{\alpha}$ from this distribution:

$$\boldsymbol{\alpha} \sim \mathcal{N}\left(MLP^\mu(\boldsymbol{x}), MLP^\Sigma(\boldsymbol{x})\right). \tag{8}$$

In practice, we implement this by utilizing the reparameterization trick [18] to obtain an unbiased estimate of the gradient and further optimize the variational distribution. The detailed derivation can be found in Appendix E.3. The minimization of the upper bound of $I(\boldsymbol{x}; \boldsymbol{\alpha})$ equals to the minimization of the Kullback-Leibler divergence between $q_\theta(\boldsymbol{\alpha}|\boldsymbol{x})$ and $r(\boldsymbol{\alpha})$. Therefore, the loss function can be represented as:

$$\mathcal{L}_{SIM} = \mathbb{E}\left[ \sum_{i=1}^{k} D_{KL}\left(q_\theta(\boldsymbol{\alpha}_i|\boldsymbol{x}_i) \| r(\boldsymbol{\alpha}_i)\right) \right]. \tag{9}$$

**Discussion.** We further provide explanation of CIB module with the information plane [8, 6] in Appendix F. It should be noted that the PIM supervises only the representative subset $\hat{\mathbf{x}}$ containing task-relevant information (selected by the similarity between image features and corresponding class-specific concepts). Meanwhile, the SIM is applied to all patches in $\mathbf{x}$, as any patch may carry information irrelevant to the task (*e.g.*, background information and stain styles). Besides, SIM cannot be directly optimized without the guidance of concept anchors (*i.e.*, PIM) due to the absence of patch-level labels. As demonstrated in the ablation study in Section 4.4, the absence of concept anchor guidance leads to the collapse of discriminative information in the calibrated feature, adversely affecting downstream task performance. By maximizing predictive information and minimizing superfluous details, the CIB module effectively enhances the discriminative capacity of the original features and aligns them with the task-specific concept anchors for improved prediction.

## 3.3 Concept-Feature Interference

We also propose the Concept-Feature Interference (CFI) module to utilize the similarity characteristic between calibrated features and concept anchors to further obtain robust and discriminative information for the downstream tasks. Our primary focus is on the class-specific concept anchors $\mathbf{c}^{\mathrm{cs}}$. Specifically, for each CIB encoded feature $\boldsymbol{\alpha}_i$, we calculate the cosine similarity between $\boldsymbol{\alpha}_i$ and each class-specific concept $\boldsymbol{c}_i^{\mathrm{cs}}$. It is important to note that the number of class-specific concepts $m$ is larger than the number of classes, as we use multiple *<CLASSNAME>* and templates to generate the concept anchor for each class, as shown in the Appendix G. Thus, we can obtain the similarity vector by concatenating the similarity scores between $\boldsymbol{\alpha}_i$ and each class-specific concept $\boldsymbol{c}_i^{\mathrm{cs}}$. To integrate the interference information (similarity relationship) into the enhanced feature, we align the similarity vector with the calibrated feature $\boldsymbol{\alpha}_i$ using a Self-Normalizing Network (SNN) layer [19]. This allows us to obtain the final interference vector $\boldsymbol{\beta}_i$ of CFI:

$$\boldsymbol{\beta}_i = \mathrm{SNN}\left(\mathrm{Concat}\left[\{\mathrm{Sim}\left(\boldsymbol{\alpha}_i, \boldsymbol{c}_i^{\mathrm{cs}}\right)\}_{i=1}^m\right]\right). \tag{10}$$

The interference vector contains superficial information that indicates the similarity between the calibrated feature and concept anchor directly. This is completely different from the calibrated feature of the CIB module, which contains discriminative latent information for the downstream tasks. Therefore, integrating the interference feature can further provide robust and discriminative information for the downstream tasks.

**Discussion.** The CFI module is designed to utilize the similarity characteristic between calibrated feature and concept anchor as a discriminative feature, which can be further integrated into the calibrated feature for downstream tasks. This is different from other studies that directly compare the similarity between visual features and textual concept features of different classes to perform zero-shot classification [25].

## 3.4 Training Objective

The overall training objective of the CATE framework can be represented as the combination of the cross entropy loss $\mathcal{L}_{CE}$ for the downstream tasks, the predictive information maximization loss $\mathcal{L}_{PIM}$, and the superfluous information minimization loss $\mathcal{L}_{SIM}$:

$$\mathcal{L} = \mathcal{L}_{CE} + \lambda_P \mathcal{L}_{PIM} + \lambda_S \mathcal{L}_{SIM}, \tag{11}$$

where $\lambda_P$ and $\lambda_S$ are hyperparameters and influence of them is discussed in Appendix C.

## 4 Experiments

### 4.1 Experimental Settings

**Tasks and Datasets.** We conducted cancer subtyping tasks on three public WSI datasets from The Cancer Genome Atlas (TCGA) project: Invasive Breast Carcinoma (BRCA), Non-Small Cell Lung Cancer (NSCLC), and Renal Cell Carcinoma (RCC). Detailed dataset information is available in Appendix D.

Table 1: Cancer Subtyping Results on BRCA of MIL Models Incorporated with CATE.

| Method | CATE | BRCA ($N_{IND}$=1) | | | | | | | |
|---|---|---|---|---|---|---|---|---|---|
| | | OOD-AUC | Gain | OOD-ACC | Gain | IND-AUC | Gain | IND-ACC | Gain |
| ABMIL | ✗ | 0.914±0.015 | N/A | 0.852±0.014 | N/A | 0.963±0.044 | N/A | 0.888±0.053 | N/A |
| CLAM | ✗ | 0.907±0.017 | N/A | 0.802±0.053 | N/A | 0.965±0.049 | N/A | 0.888±0.068 | N/A |
| DSMIL | ✗ | 0.925±0.020 | N/A | 0.836±0.048 | N/A | 0.969±0.040 | N/A | 0.900±0.080 | N/A |
| DTFD-MIL | ✗ | 0.912±0.012 | N/A | 0.858±0.020 | N/A | 0.944±0.058 | N/A | 0.894±0.070 | N/A |
| TransMIL | ✗ | 0.918±0.015 | N/A | 0.832±0.046 | N/A | 0.969±0.036 | N/A | 0.918±0.067 | N/A |
| R$^2$T-MIL[†] | ✗ | 0.901±0.027 | N/A | 0.816±0.051 | N/A | 0.965±0.033 | N/A | 0.894±0.022 | N/A |
| ABMIL | ✓ | **0.951**±0.003 | ↑4.05% | 0.897±0.026 | ↑5.28% | **0.998**±0.006 | ↑3.63% | **0.965**±0.045 | ↑8.67% |
| CLAM | ✓ | **0.951**±0.005 | ↑4.85% | **0.906**±0.020 | ↑12.97% | **0.998**±0.006 | ↑3.42% | **0.965**±0.037 | ↑8.67% |
| DSMIL | ✓ | 0.936±0.007 | ↑1.19% | 0.866±0.036 | ↑3.59% | 0.990±0.022 | ↑2.17% | 0.959±0.044 | ↑6.56% |
| DTFD-MIL | ✓ | 0.947±0.004 | ↑3.84% | **0.906**±0.009 | ↑5.59% | 0.985±0.028 | ↑4.34% | 0.953±0.042 | ↑6.60% |
| TransMIL | ✓ | 0.938±0.005 | ↑2.18% | 0.880±0.023 | ↑5.77% | **0.998**±0.006 | ↑2.99% | **0.965**±0.027 | ↑5.12% |

| Method | CATE | BRCA ($N_{IND}$=2) | | | | | | | |
|---|---|---|---|---|---|---|---|---|---|
| | | OOD-AUC | Gain | OOD-ACC | Gain | IND-AUC | Gain | IND-ACC | Gain |
| ABMIL | ✗ | 0.899±0.035 | N/A | 0.892±0.019 | N/A | 0.967±0.019 | N/A | 0.941±0.024 | N/A |
| CLAM | ✗ | 0.893±0.030 | N/A | 0.862±0.042 | N/A | 0.960±0.042 | N/A | 0.935±0.027 | N/A |
| DSMIL | ✗ | 0.881±0.032 | N/A | 0.852±0.028 | N/A | 0.946±0.057 | N/A | 0.940±0.020 | N/A |
| DTFD-MIL | ✗ | 0.909±0.019 | N/A | 0.878±0.014 | N/A | 0.973±0.023 | N/A | 0.945±0.041 | N/A |
| TransMIL | ✗ | 0.904±0.023 | N/A | 0.852±0.090 | N/A | 0.966±0.031 | N/A | 0.936±0.052 | N/A |
| R$^2$T-MIL[†] | ✗ | 0.902±0.028 | N/A | 0.873±0.027 | N/A | 0.946±0.060 | N/A | 0.929±0.048 | N/A |
| ABMIL | ✓ | 0.943±0.006 | ↑4.89% | **0.907**±0.018 | ↑1.68% | **0.981**±0.018 | ↑1.45% | 0.948±0.030 | ↑0.74% |
| CLAM | ✓ | 0.945±0.008 | ↑5.82% | 0.896±0.030 | ↑3.94% | 0.976±0.023 | ↑1.67% | 0.938±0.043 | ↑0.32% |
| DSMIL | ✓ | 0.919±0.015 | ↑4.31% | 0.869±0.036 | ↑2.00% | 0.958±0.051 | ↑1.27% | **0.949**±0.024 | ↑0.96% |
| DTFD-MIL | ✓ | **0.946**±0.005 | ↑4.07% | 0.887±0.027 | ↑1.03% | 0.977±0.023 | ↑0.41% | 0.946±0.036 | ↑0.11% |
| TransMIL | ✓ | 0.920±0.011 | ↑1.77% | 0.867±0.046 | ↑1.76% | 0.968±0.045 | ↑0.21% | 0.940±0.026 | ↑0.43% |

[*] The best results are highlighted in **bold**, and the second-best results are underlined.

[†] R$^2$T-MIL is designed for feature re-embedding that utilize ABMIL as base MIL model.

**IND and OOD Settings.** The datasets in the TCGA contains samples from different source sites (*i.e.*, different hospitals or laboratories), which are indicated in the sample barcodes[2]. And different source sites have different staining protocols and imaging characteristics, causing feature **domain shifts** between different sites [4, 5]. Therefore, MIL models trained on several sites may not generalize well to others. To better evaluate the true performance of the models, we selected several sites as IND data (in-domain, the testing and training data are from the same sites), and used data from other sites as OOD data (out-of-domain, the testing and training data are from different sites), and reported the testing performance on both IND and OOD data. Specifically, we designated $N_{IND}$ sites as IND and the remaining as OOD. Each experiment involved splitting the IND data into training, validation, and testing sets, training the models on IND data, and evaluating them on both IND and OOD testing data. For the BRCA dataset, we randomly selected one or two sites as IND data and used the remaining sites as OOD data. *However, for NSCLC (2 categories) and RCC (3 categories) datasets, each site contains samples from only one subtype.* Therefore, we cannot select only one site as IND data, as it will include one category/subtype in the training data. Instead, we randomly selected one or two corresponding sites for **each category** as IND data for NSCLC and RCC, and used the other sites as OOD data. Finally, we obtained 1 or 2 IND sites for BRCA, 2 or 4 for NSCLC, and 3 or 6 for RCC.

**Evaluation.** We report the area under the receiver operating characteristic curve (AUC) and accuracy for the OOD and IND test sets, respectively, with means and standard deviations over 10 runs of Monte-Carlo Cross Validation. Notably, the **OOD performance** is emphasized for NSCLC and RCC, where *each site contains samples from only one cancer subtype*. Traditional MIL models tend to recognize site-specific patterns (*e.g.*, staining) as **shortcuts** and excel in in-domain evaluations, rather than identifying useful class-specific features, making performance less reflective of the models' actual capability. Therefore, OOD performance more accurately reflects the models' discriminative and generalization capabilities.

**Comparisons.** Given that CATE is an adaptable method, we evaluated the performance variations across various MIL models both with and without the integration of CATE. We specifically focused on the following state-of-the-art MIL models: the original ABMIL [16], CLAM [26], DSMIL [21],

---

[2]https://docs.gdc.cancer.gov/Encyclopedia/pages/TCGA_Barcode/

Table 2: Cancer Subtyping Results on NSCLC and RCC.

| Method | NSCLC ($N_{IND}$=2) | | | | NSCLC ($N_{IND}$=4) | | | |
|---|---|---|---|---|---|---|---|---|
| | OOD-AUC | OOD-ACC | IND-AUC[#] | IND-ACC[#] | OOD-AUC | OOD-ACC | IND-AUC[#] | IND-ACC[#] |
| ABMIL | 0.874±0.021 | 0.803±0.021 | 0.997±0.004 | 0.954±0.028 | 0.951±0.023 | 0.883±0.029 | 0.974±0.018 | 0.910±0.036 |
| CLAM | 0.875±0.020 | 0.801±0.021 | 0.997±0.007 | 0.963±0.042 | 0.931±0.037 | 0.870±0.036 | 0.977±0.023 | 0.926±0.048 |
| DSMIL | 0.839±0.046 | 0.764±0.043 | 0.993±0.004 | 0.963±0.028 | 0.934±0.019 | 0.864±0.026 | 0.974±0.013 | 0.913±0.042 |
| DTFD-MIL | 0.903±0.023 | 0.836±0.026 | 0.990±0.009 | 0.958±0.049 | 0.949±0.010 | 0.893±0.012 | 0.981±0.012 | 0.918±0.040 |
| TransMIL | 0.790±0.028 | 0.712±0.024 | 0.997±0.004 | 0.954±0.033 | 0.917±0.022 | 0.832±0.031 | 0.977±0.014 | 0.923±0.029 |
| R²T-MIL † | 0.739±0.088 | 0.690±0.075 | 0.999±0.002 | 0.971±0.036 | 0.892±0.041 | 0.800±0.059 | 0.977±0.018 | 0.916±0.045 |
| **CATE-MIL** | **0.945**±0.016 | **0.840**±0.043 | 0.985±0.011 | 0.938±0.037 | **0.969**±0.003 | **0.906**±0.011 | 0.967±0.019 | 0.905±0.054 |

| Method | RCC ($N_{IND}$=3) | | | | RCC ($N_{IND}$=6) | | | |
|---|---|---|---|---|---|---|---|---|
| | OOD-AUC | OOD-ACC | IND-AUC[#] | IND-ACC[#] | OOD-AUC | OOD-ACC | IND-AUC[#] | IND-ACC[#] |
| ABMIL | 0.973±0.005 | 0.891±0.017 | 0.997±0.004 | 0.961±0.032 | 0.971±0.007 | 0.885±0.010 | 0.973±0.010 | 0.897±0.023 |
| CLAM | 0.972±0.004 | 0.893±0.012 | 0.991±0.005 | 0.961±0.032 | 0.969±0.009 | 0.888±0.015 | 0.975±0.011 | 0.896±0.031 |
| DSMIL | 0.977±0.002 | 0.893±0.010 | 0.996±0.006 | 0.965±0.026 | 0.969±0.008 | 0.883±0.016 | 0.980±0.012 | 0.901±0.022 |
| DTFD-MIL | 0.975±0.003 | 0.897±0.012 | 0.996±0.004 | 0.943±0.046 | 0.971±0.007 | 0.893±0.017 | 0.974±0.012 | 0.878±0.022 |
| TransMIL | 0.961±0.010 | 0.864±0.022 | 0.994±0.004 | 0.930±0.030 | 0.947±0.017 | 0.828±0.037 | 0.975±0.013 | 0.894±0.027 |
| R²T-MIL † | 0.956±0.018 | 0.847±0.022 | 0.991±0.008 | 0.936±0.030 | 0.932±0.020 | 0.803±0.048 | 0.974±0.012 | 0.897±0.029 |
| **CATE-MIL** | **0.983**±0.002 | **0.911**±0.018 | 0.989±0.009 | 0.944±0.031 | **0.979**±0.007 | **0.905**±0.017 | 0.963±0.011 | 0.882±0.032 |

[*] The best results are highlighted in **bold**, and the second-best results are underlined.

[†] R²T-MIL is designed for feature re-embedding that utilize ABMIL as base MIL model.

[#] The in-domain performance of NSCLC and RCC does not represent the true ability of the models, as each site contains only samples from one cancer subtype. We primarily focus on the **OOD performance** for these two datasets.

TransMIL [33], DTFD-MIL [40], and R²T-MIL [35]. The R²T-MIL [35] is a feature re-embedding method that utilizes ABMIL as the base MIL model.

**Implementation Details.** This study begins the image feature extraction process by segmenting the foreground tissue and then splitting the WSI into $512\times512$ pixels patches at $20\times$ magnification. Subsequently, these patches are processed through a pre-trained image encoder from CONCH [25] to extract image features. For concept anchors, we utilize CONCH's text encoder to derive task-relevant concepts from predefined text prompts, with detailed prompt information available in Appendix G. Model parameters are optimized using the Adam optimizer with a learning rate of $10^{-5}$. The batch size is set to 1, and all the experiments are conducted on a single NVIDIA RTX 3090 GPU.

## 4.2 Experimental Results

**Quantitative Results on BRCA Dataset.** To fully evaluate the effectiveness of CATE, we assessed its impact on several state-of-the-art MIL models using the BRCA dataset. The results are shown in Table 1, where the MIL models integrated with CATE outperform their original counterparts in both in-domain (IND) and out-of-domain (OOD) testing, which demonstrates the effectiveness and generalization capabilities of CATE. Comparing with R²T-MIL, which is a feature re-embedding method that utilizes ABMIL as the base MIL model, CATE incorporated with ABMIL consistently achieves better performance in terms of both OOD and IND testing. To further investigate the effectiveness of CATE, we conducted experiments by altering the in-domain sites and applying traditional settings. Detailed results are available in Appendix B.

**Qualitative Analysis.** To qualitatively investigate the effectiveness of CATE, we visualized attention heatmaps, UMAP, and the similarities between original features and corresponding class concept features, as well as calibrated features and class concept features, as shown in Figure 3. Additional visualization results are provided in Appendix H. As shown in Figure 3 (a&b), attention heatmap comparisons reveal that CATE-MIL focuses more intensely on cancerous regions, with a clearer delineation between high and low attention areas. By comparing the similarities of original and calibrated features to class concept features in Figure 3 (c&d), it is evident that the enhanced similarity in cancerous regions is significantly higher than in original features. Moreover, the disparity between cancerous and non-cancerous regions' similarities is also expanded, which further verifies the ability of CATE to enhance task-relevant information and suppress irrelevant information. We further performed a UMAP visualization of class concept features, original features, and calibrated features. As depicted in Figure 3 (f), calibrated features are notably closer to the corresponding class (IDC)

concept features compared to the original features, which demonstrates CATE's ability to effectively align features with task-relevant concepts and enhance task-relevant information.

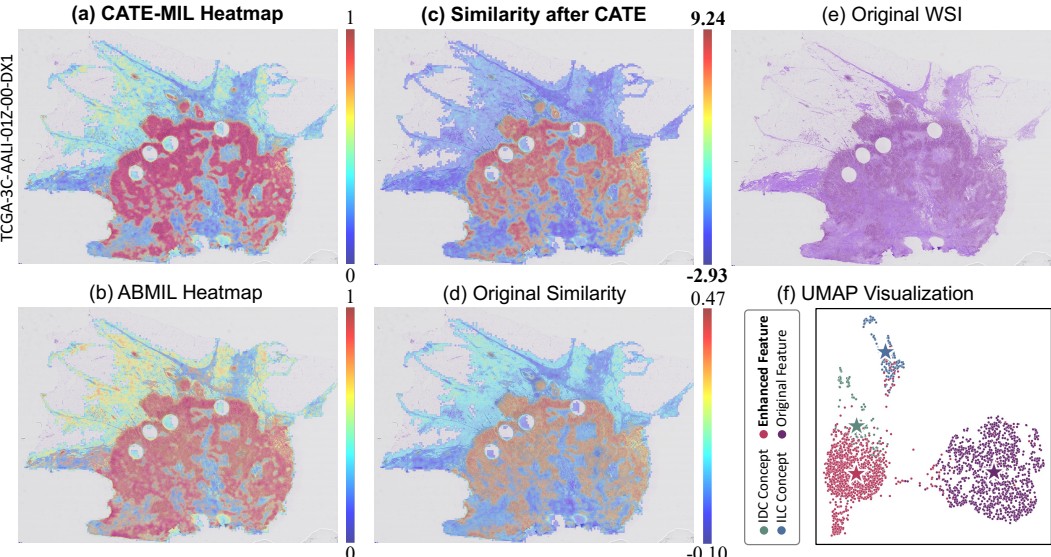

Figure 3: (a) Attention heatmap of CATE-MIL. (b) Attention heatmap of the original ABMIL. (c) similarity between the calibrated features and the corresponding class concept feature. (d) similarity between the original features and the corresponding class concept feature. (e) Original WSI. (f) UMAP visualization of class concept features, original features, and enhanced features.

### 4.3 Results on Additional Datasets

For clarity and to highlight the superiority of ABMIL when enhanced with CATE, we developed **CATE-MIL** by incorporating CATE into ABMIL and compared it against other leading MIL models on NSCLC and RCC datasets. The comparative results in Table 2 confirm that CATE-MIL consistently outperforms other models in both OOD and IND performance. However, it is noted that CATE-MIL performs poorly on the in-domain testing data for NSCLC and RCC. This underperformance may be attributed to the elimination of task-irrelevant information, including site-specific patterns, by CATE, potentially degrading performance on in-domain data for these datasets. Consequently, OOD performance more accurately reflects the discriminative and generalization capabilities of the models.

### 4.4 Ablation Analysis

We conduct ablation studies to assess the effectiveness of each component within CATE, and the results are shown in Table 3. Initially, incorporating Predictive Information Maximization (PIM) enables ABMIL to achieve improved performance in most experiments, which demonstrates PIM's efficacy in extracting task-relevant information. However, using Superfluous Information Minimization (SIM) alone results in performance degradation across most experiments, which suggests that SIM may discard some task-relevant information without guidance from a task-relevant concept anchor. Incorporating both PIM and SIM consistently enhances ABMIL's performance in all experiments, which further verifies that their combination effectively boosts the generalization capabilities of MIL models. We also conduct experiments by only using the interference features in CFI as the input of ABMIL, and the results show that the interference features are also informative for WSI classification tasks. More ablation analysis about the weights of PIM and SIM in CIB module the number of representative patches are in Appendix C.

## 5 Conclusion and Discussion

In this paper, we introduce CATE, a new approach that offers a *"free lunch"* for task-specific adaptation of pathology VLM by leveraging the inherent consistency between image and text modalities.

Table 3: Ablation study of CATE.

| Method | PIM | SIM | CFI | BRCA ($N_{\text{IND}}$=1) | BRCA ($N_{\text{IND}}$=2) | NSCLC ($N_{\text{IND}}$=2) | NSCLC ($N_{\text{IND}}$=4) | RCC ($N_{\text{IND}}$=3) | RCC ($N_{\text{IND}}$=6) |
|---|---|---|---|---|---|---|---|---|---|
| ABMIL | | | | 0.914±0.015 | 0.899±0.035 | 0.874±0.021 | 0.951±0.023 | 0.973±0.005 | 0.971±0.007 |
| | ✓ | | | 0.932±0.011 | 0.939±0.012 | 0.906±0.026 | 0.901±0.047 | 0.976±0.005 | 0.973±0.005 |
| | | ✓ | | 0.895±0.022 | 0.885±0.140 | 0.656±0.041 | 0.898±0.028 | 0.952±0.014 | 0.954±0.015 |
| | ✓ | ✓ | | 0.936±0.010 | 0.942±0.010 | 0.910±0.030 | 0.960±0.011 | 0.979±0.004 | 0.977±0.005 |
| | | | ✓ | 0.913±0.024 | 0.884±0.032 | 0.850±0.036 | 0.918±0.029 | 0.975±0.008 | 0.941±0.022 |
| **CATE-MIL** | ✓ | ✓ | ✓ | **0.951**±0.003 | **0.943**±0.006 | **0.945**±0.016 | **0.970**±0.003 | **0.983**±0.002 | **0.979**±0.007 |

CATE shows the potential to enhance the generic features extracted by pathology VLM for specific downstream tasks, using task-specific concept anchors as guidance. The proposed CIB module calibrates the image features by enhancing task-relevant information while suppressing task-irrelevant information, while the CFI module obtains the interference vector for each patch to generate discriminative task-specific features. Extensive experiments on WSI datasets demonstrate the effectiveness of CATE in improving the performance and generalizability of state-of-the-art MIL methods.

**Limitations and Social Impact.** The proposed CATE offers a promising solution to customize the pathology VLM for specific tasks, significantly improving the performance and applicability of MIL methods in WSI analysis. However, the performance of CATE heavily depends on the quality of the concept anchors, which, in turn, relies on domain knowledge and the quality of the pre-trained pathology VLM. Additionally, while CATE is optimized for classification tasks such as cancer subtyping, it may not be readily applicable to other analytical tasks, such as survival prediction. However, there might be a potential solution to address this challenge. For instance, we could leverage LLMs or retrieval-based LLMs to generate descriptive prompts about the general morphological appearance of WSIs for specific cancer types. By asking targeted questions, we can summarize reliable and general morphological descriptions associated with different survival outcomes or biomarker expressions and further verify these prompts with pathologists. Moreover, since medical data may contain sensitive information, ensuring the privacy and security of such data is crucial.

# 6    Acknowledgements

This work was supported in part by the Research Grants Council of Hong Kong (27206123 and T45-401/22-N), in part by the Hong Kong Innovation and Technology Fund (ITS/274/22), in part by the National Natural Science Foundation of China (No. 62201483), and in part by Guangdong Natural Science Fund (No. 2024A1515011875).

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

# A  Overview

The structure of this supplementary material as shown below,

- Appendix B presents additional experimental results, including changes in in-domain and out-of-domain site splitting, further comparisons on the BRCA dataset under traditional settings, additional results on the PRAD dataset, and results under site-preserved cross-validation.
- Appendix C discusses additional ablation study results concerning hyperparameters, including the impact of loss weights for PIM and SIM, the effect of the number of representative patches $k$, and further ablation studies on the CIB module.
- Appendix D presents detailed descriptions of datasets and experimental settings.
- Appendix E provides the detailed definition and formula derivation.
- Appendix F elaborates on the Concept Information Bottleneck (CIB) module using the information plane.
- Appendix G details the prompts used in our experiments.
- Appendix H presents additional visualization results.

# B  Additional Results

## B.1  Additional Cancer Subtyping Results with Different IND and OOD Sites

To further evaluate the generalization ability of the proposed CATE method, we conduct additional experiments on TCGA-BRCA dataset with different settings of in-domain (IND) and out-of-domain (OOD) sites. The detailed experimental settings are shown below:

- BRCA ($N_{\text{IND}}$=1): One site as the in-domain data, and the other sites as the out-of-domain data. The details of the in-domain site are shown below:
    - AR: 44 slides of IDC and 15 slides of ILC.
- BRCA ($N_{\text{IND}}$=2): Two sites as the in-domain data, and the other sites as the out-of-domain data. The details of the in-domain sites are shown below:
    - AR: 44 slides of IDC and 15 slides of ILC.
    - B6: 39 slides of IDC and 6 slides of ILC.

The additional results are presented in Table 4. It is evident that the proposed CATE-MIL consistently delivers superior performance in both in-domain and out-of-domain settings, demonstrating its effectiveness in enhancing task-specific information and minimizing the impact of irrelevant data.

## B.2  Additional Cancer Subtyping Results with Traditional Experimental Settings

In the main paper, we divided the dataset into in-domain and out-of-domain sites to evaluate the generalization ability of the proposed CATE method. To further evaluate its effectiveness, we conducted additional experiments under traditional settings, randomly splitting the dataset into training, validation, and testing sets. We report the performance metrics, including means and standard deviations over 10 Monte-Carlo Cross-Validation runs, in Table 5. Additionally, we provide comparisons of training times and parameter sizes for various methods.

It is evident that the proposed CATE-MIL consistently outperforms others in both AUC and ACC metrics, underscoring its superiority. Furthermore, CATE-MIL benefits from shorter training times and smaller parameter sizes compared to the $R^2$T-MIL method.

## B.3  Additional Gleason Grading Results on PRAD

CATE is a general framework that can be applied to more complex tasks beyond cancer subtyping, such as Gleason grading in prostate cancer. We have conducted conducted experiments on the TCGA-PRAD dataset to evaluate the performance of CATE-MIL in Gleason grading. Specifically,

Table 4: Supplementary Cancer Subtyping Results on BRCA.

| Method | BRCA ($N_{\text{IND}}$=1) | | | |
| --- | --- | --- | --- | --- |
| | OOD-AUC | OOD-ACC | IND-AUC | IND-ACC |
| ABMIL | $0.916 \pm 0.017$ | $\underline{0.882} \pm 0.022$ | $0.977 \pm 0.029$ | $0.953 \pm 0.036$ |
| CLAM | $0.895 \pm 0.031$ | $0.842 \pm 0.049$ | $0.973 \pm 0.035$ | $0.938 \pm 0.043$ |
| DSMIL | $0.892 \pm 0.017$ | $0.863 \pm 0.016$ | $0.967 \pm 0.022$ | $0.924 \pm 0.048$ |
| TransMIL | $0.917 \pm 0.008$ | $0.878 \pm 0.016$ | $0.980 \pm 0.031$ | $0.955 \pm 0.038$ |
| DTFD-MIL | $\underline{0.920} \pm 0.023$ | $0.881 \pm 0.024$ | $\underline{0.982} \pm 0.030$ | $\underline{0.957} \pm 0.038$ |
| [†]R$^2$T-MIL | $0.902 \pm 0.022$ | $0.835 \pm 0.063$ | $0.963 \pm 0.037$ | $0.944 \pm 0.039$ |
| [†]**CATE-MIL** | $\mathbf{0.938} \pm 0.014$ | $\mathbf{0.895} \pm 0.021$ | $\mathbf{0.984} \pm 0.021$ | $\mathbf{0.959} \pm 0.038$ |
| Method | BRCA ($N_{\text{IND}}$=2) | | | |
| | OOD-AUC | OOD-ACC | IND-AUC | IND-ACC |
| ABMIL | $0.914 \pm 0.012$ | $0.879 \pm 0.017$ | $0.954 \pm 0.035$ | $0.910 \pm 0.043$ |
| CLAM | $0.914 \pm 0.021$ | $0.890 \pm 0.013$ | $0.958 \pm 0.034$ | $\underline{0.926} \pm 0.050$ |
| DSMIL | $\underline{0.931} \pm 0.007$ | $\underline{0.900} \pm 0.012$ | $\underline{0.971} \pm 0.020$ | $0.919 \pm 0.034$ |
| TransMIL | $0.929 \pm 0.009$ | $0.897 \pm 0.006$ | $0.962 \pm 0.030$ | $0.903 \pm 0.039$ |
| DTFD-MIL | $0.898 \pm 0.009$ | $0.868 \pm 0.017$ | $0.950 \pm 0.034$ | $0.890 \pm 0.037$ |
| [†]R$^2$T-MIL | $0.919 \pm 0.017$ | $0.893 \pm 0.017$ | $0.962 \pm 0.025$ | $0.906 \pm 0.026$ |
| [†]**CATE-MIL** | $\mathbf{0.947} \pm 0.005$ | $\mathbf{0.920} \pm 0.004$ | $\mathbf{0.980} \pm 0.015$ | $\mathbf{0.939} \pm 0.022$ |

The best results are highlighted in **bold**, and the second-best results are underlined.
[†] denotes the methods for feature re-embedding that utilize ABMIL as base MIL model.

Table 5: Results on BRCA under Traditional Settings.

| Method | AUC | ACC | Training Time | Params Size |
| --- | --- | --- | --- | --- |
| ABMIL | $0.922_{\pm 0.046}$ | $0.902_{\pm 0.039}$ | 67.16 S | 1.26 MB |
| CLAM | $0.928_{\pm 0.030}$ | $\underline{0.912}_{\pm 0.034}$ | 67.99 S | 2.01 MB |
| DSMIL | $0.934_{\pm 0.039}$ | $0.910_{\pm 0.033}$ | 72.91 S | 1.26 MB |
| TransMIL | $0.927_{\pm 0.046}$ | $0.906_{\pm 0.032}$ | 72.39 S | 5.39 MB |
| DTFD-MIL | $\underline{0.940}_{\pm 0.030}$ | $0.901_{\pm 0.041}$ | 70.99 S | 8.03 MB |
| [†]R$^2$T-MIL | $0.936_{\pm 0.027}$ | $0.903_{\pm 0.028}$ | 68.87 S | 9.28 MB (1.26+8.02) |
| [†]**CATE-MIL** | $\mathbf{0.945}_{\pm 0.033}$ | $\mathbf{0.917}_{\pm 0.029}$ | 68.53 S | 4.26 MB (1.26+3.00) |

The best results are highlighted in **bold**, and the second-best results are underlined.
[†] denotes the methods for feature re-embedding that utilize ABMIL as base MIL model.

Table 6: Supplementary Gleason Grading Results on PRAD.

| Method | PRAD | | | |
| --- | --- | --- | --- | --- |
| | OOD-AUC | OOD-ACC | IND-AUC | IND-ACC |
| ABMIL | $0.704 \pm 0.034$ | $0.510 \pm 0.075$ | $0.742 \pm 0.060$ | $0.575 \pm 0.051$ |
| **CATE-MIL** | $\mathbf{0.755} \pm 0.050$ | $\mathbf{0.567} \pm 0.067$ | $\mathbf{0.797} \pm 0.044$ | $\mathbf{0.643} \pm 0.075$ |

Table 7: Supplementary Results under Site-Preserved Cross-Validation.

| Dataset | Method | OOD-AUC | IND-AUC |
| --- | --- | --- | --- |
| BRCA | ABMIL | $0.912 \pm 0.012$ | $0.905 \pm 0.043$ |
| | **CATE-MIL** | $\mathbf{0.935} \pm 0.014$ | $\mathbf{0.942} \pm 0.038$ |
| NSCLC | ABMIL | $0.942 \pm 0.016$ | $0.941 \pm 0.013$ |
| | **CATE-MIL** | $\mathbf{0.951} \pm 0.015$ | $\mathbf{0.943} \pm 0.008$ |
| RCC | ABMIL | $0.980 \pm 0.001$ | $0.986 \pm 0.010$ |
| | **CATE-MIL** | $\mathbf{0.983} \pm 0.001$ | $\mathbf{0.989} \pm 0.007$ |

the samples in PRAD dataset are divided into Gleason pattern 3, 4, and 5, and the task is to classify the samples into these three categories. The results are shown in Table 6. It is evident that CATE-MIL consistently outperforms the base model ABMIL in both in-domain and out-of-domain settings, demonstrating its effectiveness in enhancing task-specific information and minimizing the impact of irrelevant data. In the future, as more studies reveal the connection between morphological features and molecular biomarkers and more powerful pathology VLMs are developed, our framework has the potential to benefit more complex tasks.

### B.4 Additional Experiments using Site-Preserved Cross-Validation

To provide a more comprehensive evaluation of the proposed CATE-MIL, we conduct additional experiments using site-preserved cross-validation [11], where the samples from the same site are preserved in the same fold. For each fold, we split the data into training and testing sets, and these testing sets are regarded as in-domain testing data. And the other sites are used as out-of-domain testing data. The results are shown in Table 7. It is evident that CATE-MIL consistently outperforms ABMIL in both in-domain and out-of-domain settings.

## C   Additional Ablation Study

### C.1   Ablation Study of the Weight of PIM and SIM Losses

The overall objective of the proposed CATE-MIL is a weighted sum of the PIM and SIM losses, along with the classification loss:

$$\mathcal{L} = \mathcal{L}_{CE} + \lambda_P \mathcal{L}_{PIM} + \lambda_S \mathcal{L}_{SIM}. \tag{12}$$

We note that the hyperparameters for the weights of the PIM and SIM losses were not specifically tuned in the main paper. To investigate the effect of the weight of PIM and SIM losses on the model performance, we conduct an ablation study on BRCA ($N_{\text{IND}}$=2) with CATE-MIL (ABMIL integrated with CATE), varying the weights of PIM and SIM losses. The results are shown in Figure 4.

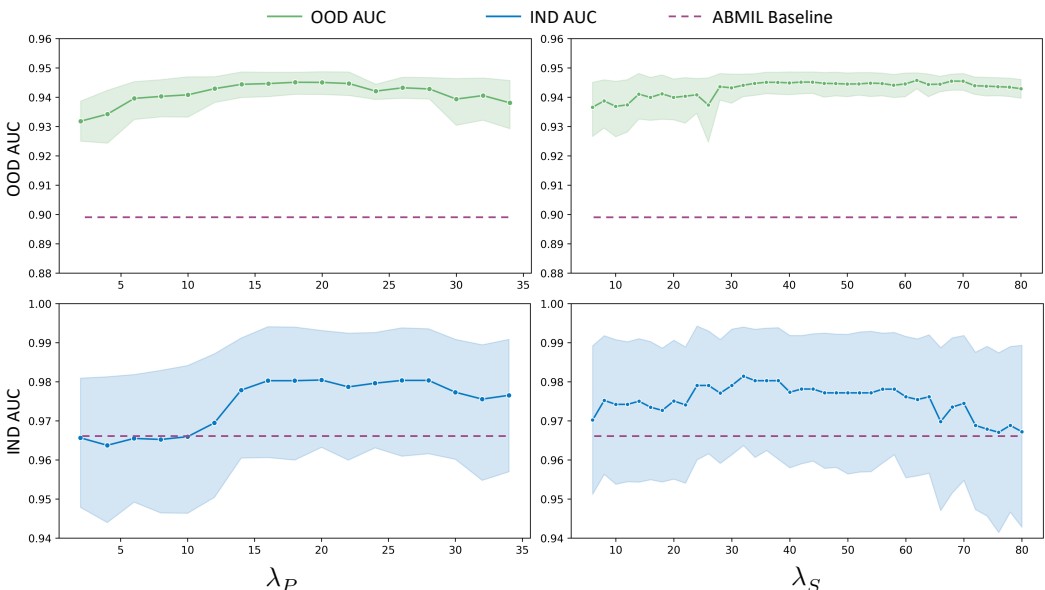

Figure 4:   Ablation study of the weight of PIM and SIM losses on the model performance.

From Figure 4, it is evident that CATE enhances the performance of the base model ABMIL in most scenarios, particularly in out-of-domain settings. When the weight of PIM loss $\lambda_P$ is too low, model performance suffers, underscoring the significant role of PIM loss in enhancing task-specific information. Conversely, an excessively high $\lambda_P$ also diminishes performance, as the model overly

prioritizes maximizing mutual information between the original and calibrated features at the expense of optimizing classification loss. Regarding the weight of SIM loss $\lambda_S$, optimal performance is achieved when $\lambda_S$ is approximately 30. If the $\lambda_S$ is too low, the model fails to effectively eliminate irrelevant information, thereby impairing performance. Conversely, if $\lambda_S$ is too high, the model risks discarding task-relevant information, leading to performance degradation.

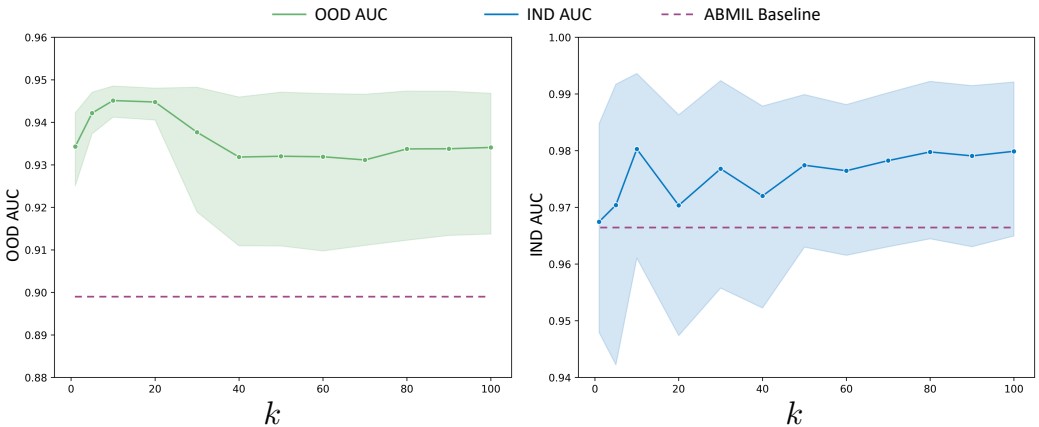

Figure 5: Ablation study of $k$.

## C.2 Ablation Study of the Number of Representative Patches $k$

In this section, we conduct an ablation study to investigate the impact of the number of representative patches, $k$, on model performance, as discussed in Section 3.2. In practice, the representative patches are selected based on the similarities between the original image feature and the corresponding class-specific concept anchor.

As shown in Figure 5, it is evident that CATE generally enhances the performance of the base model, ABMIL. When the number of representative patches $k$ is too small, model performance degrades due to insufficient capture of task-specific information. Conversely, an excessively large $k$ also leads to performance degradation in out-of-domain scenarios, as it introduces noise from irrelevant information. Consequently, we have set the number of representative patches $k$ to 10 in the main paper.

## C.3 Additional Ablation Study of CIB Module

We further conducted experiments on CATE-MIL without concept alignment (discarding PIM loss and SIM loss of the CIB module) and replaced the CIB module with an MLP to investigate the effect of concept alignment and the increased number of parameters. The results are shown in Table 8. The performance of CATE-MIL significantly decreases in both cases, demonstrating the importance of concept alignment in the CIB module and that the improvements of CATE are not due to the increased number of parameters.

Table 8: Supplementary Ablation Study of CIB Module.

| Method | BRCA ($N_{\text{IND}}=1$) | BRCA ($N_{\text{IND}}=2$) | NSCLC ($N_{\text{IND}}=2$) | NSCLC ($N_{\text{IND}}=4$) | RCC ($N_{\text{IND}}=3$) | RCC ($N_{\text{IND}}=6$) |
|---|---|---|---|---|---|---|
| **CATE-MIL** w/o CFI (Baseline) | **0.936**±0.010 | **0.942**±0.010 | **0.910**±0.030 | **0.960**±0.011 | **0.979**±0.004 | **0.977**±0.005 |
| - w/o concept alignment | 0.900±0.017 | 0.884±0.033 | 0.742±0.059 | 0.897±0.022 | 0.961±0.011 | 0.932±0.016 |
| - Replace CIB with MLP | 0.888±0.027 | 0.902±0.037 | 0.816±0.040 | 0.931±0.023 | 0.966±0.006 | 0.951±0.021 |

## D    Datasets Description and Detailed Experimental Settings

In this section, we provide detailed descriptions of the datasets used in the experiments and the detailed experimental settings that we used in the experiments in Section 4.2.

### D.1 Datasets Description

- **TCGA-BRCA**: The dataset contains nine disease subtypes, and this study focuses on the classification of Invasive ductal carcinoma (IDC, 726 slides from 694 cases) and invasive lobular carcinoma (ILC, 149 slides from 143 cases). The dataset is collected from 36 sites with 20 of them having both IDC and ILC slides, and the other 16 sites only have IDC slides or ILC slides.
- **TCGA-NSCLC**: The dataset contains two disease subtypes, including lung adenocarcinoma (LUAD, 492 slides from 430 cases) and lung squamous cell carcinoma (LUSC, 466 slides from 432 cases). The dataset is collected from 66 sites. Different from TCGA-BRCA, **each site only contains one disease subtype**.
- **TCGA-RCC**: The dataset contains three disease subtypes, including clear cell renal cell carcinoma (CCRCC, 498 slides from 492 cases), papillary renal cell carcinoma (PRCC, 289 slides from 267 cases), and chromophobe renal cell carcinoma (CHRCC, 118 slides from 107 cases). The dataset is collected from 55 sites. Similar to TCGA-NSCLC, **each site only contains one disease subtype**.

### D.2 Detailed Experimental Settings

To validate that the proposed CATE effectively improves MIL model performance by enhancing the task-specific information and eliminating the disturbance of task-irrelevant information, we conduct extensive experiments across three datasets under various in-domain and out-of-domain settings.

Specifically, for TCGA-BRCA dataset, we conduct the following experiments:

- BRCA ($N_{\text{IND}}$=1): One site as the in-domain data, and the other sites as the out-of-domain data. The details of the in-domain site are shown below:
  - D8: 59 slides of IDC and 8 slides of ILC.
- BRCA ($N_{\text{IND}}$=2): Two sites as the in-domain data, and the other sites as the out-of-domain data. The details of the in-domain sites are shown below:
  - A8: 48 slides of IDC and 5 slides of ILC.
  - D8: 59 slides of IDC and 8 slides of ILC.

For TCGA-NSCLC dataset, we conduct the following experiments:

- NSCLC ($N_{\text{IND}}$=2): Two sites as the in-domain data, and the other sites as the out-of-domain data. The details of the in-domain sites are shown below:
  - 44: 43 slides of LUAD and 0 slides of LUSC.
  - 22: 0 slides of LUAD and 36 slides of LUSC.
- NSCLC ($N_{\text{IND}}$=4): Four sites as the in-domain data, and the other sites as the out-of-domain data. The details of the in-domain sites are shown below:
  - 44: 43 slides of LUAD and 0 slides of LUSC.
  - 50: 20 slides of LUAD and 0 slides of LUSC.
  - 22: 0 slides of LUAD and 36 slides of LUSC.
  - 56: 0 slides of LUAD and 35 slides of LUSC.

For TCGA-RCC dataset, we conduct the following experiments:

- RCC ($N_{\text{IND}}$=3): Three sites as the in-domain data, and the other sites as the out-of-domain data. The details of the in-domain sites are shown below:
  - A3: 48 slides of CCRCC, 0 slides of CHRCC, and 0 slides of PRCC.
  - KL: 0 slides of CCRCC, 24 slides of CHRCC, and 0 slides of PRCC.
  - 2Z: 0 slides of CCRCC, 0 slides of CHRCC, and 23 slides of PRCC.
- RCC ($N_{\text{IND}}$=6): Six sites as the in-domain data, and the other sites as the out-of-domain data. The details of the in-domain sites are shown below:
  - A3: 48 slides of CCRCC, 0 slides of CHRCC, and 0 slides of PRCC.

- AK: 15 slides of CCRCC, 0 slides of CHRCC, and 0 slides of PRCC.

- KL: 0 slides of CCRCC, 24 slides of CHRCC, and 0 slides of PRCC.

- KM: 0 slides of CCRCC, 20 slides of CHRCC, and 0 slides of PRCC.

- 2Z: 0 slides of CCRCC, 0 slides of CHRCC, and 23 slides of PRCC.

- 5P: 0 slides of CCRCC, 0 slides of CHRCC, and 15 slides of PRCC.

# E   Detailed Definition and Formula Derivation

## E.1   Definition of Sufficiency and Consistency

In this part, we will define the *sufficiency* and *consistency* in the context of the concept information bottleneck (CIB) module.

First, we define the mutual information between two random variables $X$ and $Y$ as:

$$I(X;Y) = \sum p(x,y) \log \frac{p(x,y)}{p(x)p(y)}. \tag{13}$$

**Sufficiency:** To quantify the requirement that the calibrated features should be maximally informative about the label information $\mathbf{y}$, we define the sufficiency as the mutual information between the label information and the calibrated features:

*Sufficiency.* $\hat{\boldsymbol{\alpha}}$ is sufficient for $\mathbf{y} \iff I(\hat{\boldsymbol{x}}; \mathbf{y}|\hat{\boldsymbol{\alpha}}) = 0 \iff I(\hat{\boldsymbol{x}}; \mathbf{y}) = I(\hat{\boldsymbol{\alpha}}; \mathbf{y})$.

The sufficiency definition requires that the calibrated features $\hat{\boldsymbol{\alpha}}$ encapsulate all information about the label information $\mathbf{y}$ that is accessible from the original features $\hat{\boldsymbol{x}}$. In essence, the calibrated feature $\hat{\boldsymbol{\alpha}}$ of original feature $\hat{\boldsymbol{x}}$ is sufficient for determining label $\mathbf{y}$ if and only if the amount of information regarding the specific task is unchanged after the transformation.

**Consistency:** Since the image feature $\hat{\boldsymbol{x}}$ and the concept anchor $\mathbf{c}$ are consistent in the pathology VLM, we posit that any representation containing all information accessible from both the original feature and the concept anchor also encompasses the necessary discriminative label information. Thus, we define the consistency between the concept anchor and the original feature as:

*Consistency.* $\mathbf{c}$ is consistent with $\hat{\boldsymbol{x}}$ for $\mathbf{y} \iff I(\mathbf{y}; \hat{\boldsymbol{x}}|\mathbf{c}) = 0$.

The consistency definition requires that the concept anchor $\mathbf{c}$ and the original feature $\hat{\boldsymbol{x}}$ should be consistent with the label information $\mathbf{y}$.

## E.2   Maximize Predictive Information with InfoNCE

In this part, we will prove that the predictive information for the concept anchor can be maximized by maximizing the InfoNCE mutual information lower bound, which is defined in Equation 4. As in [27], the optimal value for $f(\boldsymbol{c}, \hat{\boldsymbol{\alpha}}_i)$ should be proportional to the density ratio:

$$f(\boldsymbol{c}, \hat{\boldsymbol{\alpha}}_i) \propto \frac{p(\boldsymbol{c}, \hat{\boldsymbol{\alpha}}_i)}{p(\boldsymbol{c})\,p(\hat{\boldsymbol{\alpha}}_i)}. \tag{14}$$

And we can get:

$$
\begin{aligned}
I_{NCE}^{opt}\left(\hat{\boldsymbol{\alpha}};\mathbf{c}\right) &= \underset{\hat{\boldsymbol{\alpha}},\boldsymbol{c}_{\mathrm{pos}}^{\mathrm{cs}}}{\mathbb{E}}\left[\sum_{i=1}^{k}\log\frac{\dfrac{p\left(\boldsymbol{c}_{\mathrm{pos}}^{\mathrm{cs}},\hat{\boldsymbol{\alpha}}_i\right)}{p\left(\boldsymbol{c}_{\mathrm{pos}}^{\mathrm{cs}}\right)p(\hat{\boldsymbol{\alpha}}_i)}}{\sum_{j=1}^{m}\frac{p\left(\boldsymbol{c}_j^{\mathrm{cs}},\hat{\boldsymbol{\alpha}}_i\right)}{p\left(\boldsymbol{c}_j^{\mathrm{cs}}\right)p(\hat{\boldsymbol{\alpha}}_i)}+\sum_{j=1}^{n}\frac{p\left(\boldsymbol{c}_j^{\mathrm{ca}},\hat{\boldsymbol{\alpha}}_i\right)}{p\left(\boldsymbol{c}_j^{\mathrm{ca}}\right)p(\hat{\boldsymbol{\alpha}}_i)}}\right] \\
&= -\underset{\hat{\boldsymbol{\alpha}},\boldsymbol{c}_{\mathrm{pos}}^{\mathrm{cs}}}{\mathbb{E}}\left[\sum_{i=1}^{k}\log\left(1+\frac{p\left(\boldsymbol{c}_{\mathrm{pos}}^{\mathrm{cs}}\right)p\left(\hat{\boldsymbol{\alpha}}_i\right)}{p\left(\boldsymbol{c}_{\mathrm{pos}}^{\mathrm{cs}},\hat{\boldsymbol{\alpha}}_i\right)}\left(\sum_{j\neq\mathrm{pos}}\frac{p\left(\boldsymbol{c}_j^{\mathrm{cs}},\hat{\boldsymbol{\alpha}}_i\right)}{p\left(\boldsymbol{c}_j^{\mathrm{cs}}\right)p\left(\hat{\boldsymbol{\alpha}}_i\right)}+\sum_{j=1}^{n}\frac{p\left(\boldsymbol{c}_j^{\mathrm{ca}},\hat{\boldsymbol{\alpha}}_i\right)}{p\left(\boldsymbol{c}_j^{\mathrm{ca}}\right)p\left(\hat{\boldsymbol{\alpha}}_i\right)}\right)\right)\right] \\
&\approx -\underset{\hat{\boldsymbol{\alpha}},\boldsymbol{c}_{\mathrm{pos}}^{\mathrm{cs}}}{\mathbb{E}}\left[\sum_{i=1}^{k}\log\left(1+\frac{p\left(\boldsymbol{c}_{\mathrm{pos}}^{\mathrm{cs}}\right)p\left(\hat{\boldsymbol{\alpha}}_i\right)}{p\left(\boldsymbol{c}_{\mathrm{pos}}^{\mathrm{cs}},\hat{\boldsymbol{\alpha}}_i\right)}\left((m-1)\underset{\hat{\boldsymbol{\alpha}}_i}{\mathbb{E}}\frac{p\left(\boldsymbol{c}_j^{\mathrm{cs}},\hat{\boldsymbol{\alpha}}_i\right)}{p\left(\boldsymbol{c}_j^{\mathrm{cs}}\right)p\left(\hat{\boldsymbol{\alpha}}_i\right)}+n\underset{\hat{\boldsymbol{\alpha}}_i}{\mathbb{E}}\frac{p\left(\boldsymbol{c}_j^{\mathrm{ca}},\hat{\boldsymbol{\alpha}}_i\right)}{p\left(\boldsymbol{c}_j^{\mathrm{ca}}\right)p\left(\hat{\boldsymbol{\alpha}}_i\right)}\right)\right)\right] \\
&= -\underset{\hat{\boldsymbol{\alpha}},\boldsymbol{c}_{\mathrm{pos}}^{\mathrm{cs}}}{\mathbb{E}}\left[\sum_{i=1}^{k}\log\left(1+\frac{p\left(\boldsymbol{c}_{\mathrm{pos}}^{\mathrm{cs}}\right)p\left(\hat{\boldsymbol{\alpha}}_i\right)}{p\left(\boldsymbol{c}_{\mathrm{pos}}^{\mathrm{cs}},\hat{\boldsymbol{\alpha}}_i\right)}\left(m+n-1\right)\right)\right] \\
&\leq -\underset{\hat{\boldsymbol{\alpha}},\boldsymbol{c}_{\mathrm{pos}}^{\mathrm{cs}}}{\mathbb{E}}\left[\sum_{i=1}^{k}\log\left(\frac{p\left(\boldsymbol{c}_{\mathrm{pos}}^{\mathrm{cs}}\right)p\left(\hat{\boldsymbol{\alpha}}_i\right)}{p\left(\boldsymbol{c}_{\mathrm{pos}}^{\mathrm{cs}},\hat{\boldsymbol{\alpha}}_i\right)}\left(m+n\right)\right)\right] \\
&= \sum_{i=1}^{k}I\left(\boldsymbol{c}_{\mathrm{pos}}^{\mathrm{cs}};\hat{\boldsymbol{\alpha}}_i\right)-k\log\left(m+n\right) \\
&= I\left(\boldsymbol{c}_{\mathrm{pos}}^{\mathrm{cs}};\hat{\boldsymbol{\alpha}}\right)-k\log\left(m+n\right),
\end{aligned}
\tag{15}
$$

where $C_{\mathrm{neg}}$ denotes the negative concepts, including both negative class concepts and other type concepts. We can see that the InfoNCE estimation is a lower bound of the mutual information between the concept anchor and the calibrated features:

$$
I\left(\boldsymbol{c}_{\mathrm{pos}}^{\mathrm{cs}};\hat{\boldsymbol{\alpha}}\right)\geq I_{NCE}^{opt}\left(\hat{\boldsymbol{\alpha}};\mathbf{c}\right)+k\log\left(m+n\right)\geq I_{NCE}\left(\hat{\boldsymbol{\alpha}};\mathbf{c}\right).
\tag{16}
$$

Thus, the predictive information of calibrated features for the concept anchor can be maximized by maximizing the InfoNCE mutual information lower bound.

### E.3 Superfluous Information Minimization

As shown in Equation 3, the mutual information between the original features and the calibrated features $I(\hat{\boldsymbol{x}};\hat{\boldsymbol{\alpha}})$ can be decomposed into the mutual information $I(\hat{\boldsymbol{\alpha}};\mathbf{c})$ between the calibrated features and the concept anchor and the mutual information $I(\hat{\boldsymbol{x}};\hat{\boldsymbol{\alpha}}|\mathbf{c})$ between the concept anchor and the calibrated features:

$$
I(\hat{\boldsymbol{x}};\hat{\boldsymbol{\alpha}}) = \underbrace{I(\hat{\boldsymbol{\alpha}};\mathbf{c})}_{\text{Predictive information for }\mathbf{c}} + \underbrace{I(\hat{\boldsymbol{x}};\hat{\boldsymbol{\alpha}}|\mathbf{c})}_{\text{Superfluous information}}.
\tag{17}
$$

As introduced in section 3.2, the first item is the predictive information for the concept anchor, which can be maximized by maximizing the InfoNCE mutual information lower bound. Thus, the second superfluous information item can be minimized by minimizing $I(\hat{\boldsymbol{x}};\hat{\boldsymbol{\alpha}})$. In practice, the superfluous information minimization is conducted on all patches in the subset $\mathbf{x}$, since the task-irrelevant information is distributed across all patches. Thus, the superfluous information can be minimized by minimizing $I(\boldsymbol{x};\boldsymbol{\alpha})$. As in [1], this mutual information can be represented as:

$$
\begin{aligned}
I(\boldsymbol{x};\boldsymbol{\alpha}) &= \int p\left(\boldsymbol{x},\boldsymbol{\alpha}\right)\log\frac{p\left(\boldsymbol{\alpha}|\boldsymbol{x}\right)}{p\left(\boldsymbol{\alpha}\right)}d\boldsymbol{x}d\boldsymbol{\alpha} \\
&= \int p\left(\boldsymbol{x},\boldsymbol{\alpha}\right)\log p\left(\boldsymbol{\alpha}|\boldsymbol{x}\right)d\boldsymbol{x}d\boldsymbol{\alpha}-\int p\left(\boldsymbol{\alpha}\right)\log p\left(\boldsymbol{\alpha}\right)d\boldsymbol{\alpha}.
\end{aligned}
\tag{18}
$$

However, the computing of the marginal distribution $p\left(\boldsymbol{\alpha}\right)=\int p\left(\boldsymbol{\alpha}|\boldsymbol{x}\right)p\left(\boldsymbol{x}\right)d\boldsymbol{x}$ is intractable. Thus, we can let the Gaussian distribution $r\left(\boldsymbol{\alpha}\right)$ approximate the marginal distribution $p\left(\boldsymbol{\alpha}\right)$. Since the Kullback-Leibler divergence between two distributions is non-negative, we can get:

$$
\int p\left(\boldsymbol{\alpha}\right)\log p\left(\boldsymbol{\alpha}\right)d\boldsymbol{\alpha}\geq\int p\left(\boldsymbol{\alpha}\right)\log r\left(\boldsymbol{\alpha}\right)d\boldsymbol{\alpha}.
\tag{19}
$$

Thus, the mutual information $I(\boldsymbol{x}; \boldsymbol{\alpha})$ have the upper bound:

$$
\begin{aligned}
I(\boldsymbol{x}; \boldsymbol{\alpha}) &\leq \int p(\boldsymbol{x}, \boldsymbol{\alpha}) \log p(\boldsymbol{\alpha}|\boldsymbol{x}) \, d\boldsymbol{x} d\boldsymbol{\alpha} - \int p(\boldsymbol{\alpha}) \log r(\boldsymbol{\alpha}) \, d\boldsymbol{\alpha} \\
&= \int p(\boldsymbol{x}) p(\boldsymbol{\alpha}|\boldsymbol{x}) \log \frac{p(\boldsymbol{\alpha}|\boldsymbol{x})}{r(\boldsymbol{\alpha})} \, d\boldsymbol{x} d\boldsymbol{\alpha}.
\end{aligned}
\tag{20}
$$

In practice, to compute this upper bound, we approximate distribution $p(\boldsymbol{x})$ with the empirical distribution:

$$
p(\boldsymbol{x}) = \frac{1}{N} \sum_{i=1}^{N} \delta_{\boldsymbol{x}_i}(\boldsymbol{x}).
\tag{21}
$$

Thus, the upper bound of the mutual information $I(\boldsymbol{x}; \boldsymbol{\alpha})$ can be computed as:

$$
\int p(\boldsymbol{x}) p(\boldsymbol{\alpha}|\boldsymbol{x}) \log \frac{p(\boldsymbol{\alpha}|\boldsymbol{x})}{r(\boldsymbol{\alpha})} \, d\boldsymbol{x} d\boldsymbol{\alpha} \approx \frac{1}{N} \sum_{i=1}^{N} q_\theta(\boldsymbol{\alpha}_i|\boldsymbol{x}_i) \log \frac{q_\theta(\boldsymbol{\alpha}_i|\boldsymbol{x}_i)}{r(\boldsymbol{\alpha}_i)},
\tag{22}
$$

where $q_\theta(\boldsymbol{\alpha}|\boldsymbol{x})$ is a variational distribution with parameter $\theta$ to approximate $p(\boldsymbol{\alpha}|\boldsymbol{x})$, and we implement this variational distribution with MLP:

$$
q_\theta(\boldsymbol{\alpha}|\boldsymbol{x}) = \mathcal{N}\left(\boldsymbol{\alpha}|MLP^\mu(\boldsymbol{x}), MLP^\Sigma(\boldsymbol{x})\right),
\tag{23}
$$

where $MLP^\mu$ and $MLP^\Sigma$ are implemented with MLPs which output the mean $\mu$ and covariance matrix $\Sigma$ of the Gaussian distribution. In practice, we utilize the reparameterization trick [18] to sample from the Gaussian distribution to get an unbiased estimate of the gradient to optimize the opjective. Thus, the upper bound can be optimized by minimizing the KL divergence between the variational distribution $q_\theta(\boldsymbol{\alpha}|\boldsymbol{x})$ and the Gaussian distribution $r(\boldsymbol{\alpha})$:

$$
\min_\theta \mathbb{E}_{\boldsymbol{x}}\left[D_{KL}\left(q_\theta(\boldsymbol{\alpha}|\boldsymbol{x}) \| r(\boldsymbol{\alpha})\right)\right].
\tag{24}
$$

## F   Explanation of CIB with Information Plane

To better understand the concept information bottleneck (CIB) method, we provide an explanation of CIB with the information plane [8, 6].

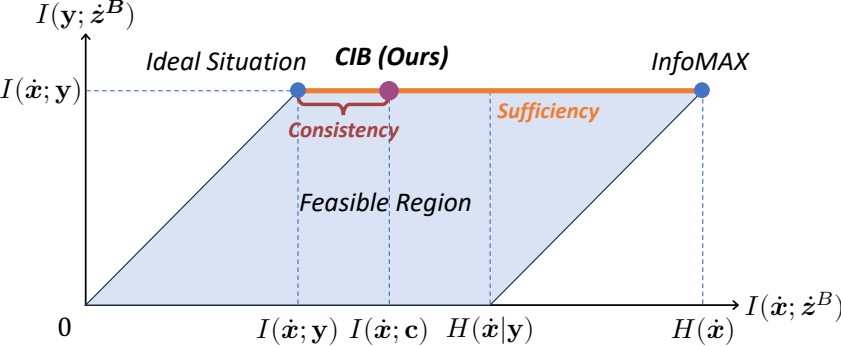

Figure 6: Explanation of CIB module with Information Plane.

As shown in Figure 6, the information plane is a two-dimensional space where the x-axis represents the mutual information between the original features and the calibrated features of CIB module $I(\hat{\boldsymbol{x}}; \hat{\boldsymbol{\alpha}})$, and the y-axis represents the mutual information between the label information and the calibrated features $I(\mathbf{y}; \hat{\boldsymbol{x}})$. The line colored orange in the information plane defines *sufficiency*, indicating where the calibrated features are maximally informative about the label information $\mathbf{y}$. The *consistency* definition falls to the left of the sufficiency line, denoting a state where the calibrated features, containing all information from both the original feature and the concept anchor, also include the label information. The InfoMAX principle [9] states that the optimal calibrated feature should be maximally informative about the original feature.

Ideally, a good calibrated feature $\hat{\alpha}$ should be maximally informative about the label information $\mathbf{y}$ (the sufficiency line in the information plane) while minimally informative about the original features $\hat{x}$ (*i.e.*, the ideal situation in the information plane). This situation can be achieved by directly supervised learning. However, the weakly supervised nature of MIL methods complicates this goal. As demonstrated in the ablation analysis in Section 4.4, if we directly perform superfluous information minimization without the guidance of concept anchor (*i.e.*, predictive information maximization), the calibrated features will be less informative about the label information $\mathbf{y}$ and the model performance will be degraded significantly. To address this issue, the proposed CIB module introduces the concept anchor $\mathbf{c}$ to guide the learning process of information bottleneck for each instance. As shown in Figure 6, the CIB module could effectively reduce the mutual information between the original features and the calibrated features $I(\hat{x}; \hat{\alpha})$ while preserving the mutual information between the label information and the calibrated features $I(\mathbf{y}; \hat{\alpha})$.

# G   Prompts

In this section, we provide the detailed prompt templates and concepts for the datasets, which are used to generate the concept anchor for the CIB module. The prompt templates are shown in Table 9, and the concepts for TCGA-BRCA, TCGA-NSCLC, and TCGA-RCC are shown in Table 10, Table 11, and Table 12, respectively. The templates and the class-agnostic prompts are referred from the original paper of CONCH [25], and the class-specific prompts are generated by querying the LLM with the question such as 'In addition to tumor tissues, what types of tissue or cells are present in whole slide images of breast cancer?' The quality of LLM generated prompts has been demonstrated in several recent studies [29, 22]. In principle, we can use LLM (*e.g.*, GPT-4) to generate reliable expert-designed prompts and further verified by pathologists. This strategy can ensure the scalability and reliability of the prompts.

Table 9: Prompt Templates.

| Templates |
| --- |
| *<CLASSNAME>*. |
| a photomicrograph showing *<CLASSNAME>*. |
| a photomicrograph of *<CLASSNAME>*. |
| an image of *<CLASSNAME>*. |
| an image showing *<CLASSNAME>*. |
| an example of *<CLASSNAME>*. |
| *<CLASSNAME>* is shown. |
| this is *<CLASSNAME>*. |
| there is *<CLASSNAME>*. |
| a histopathological image showing *<CLASSNAME>*. |
| a histopathological image of *<CLASSNAME>*. |
| a histopathological photograph of *<CLASSNAME>*. |
| a histopathological photograph showing *<CLASSNAME>*. |
| shows *<CLASSNAME>*. |
| presence of *<CLASSNAME>*. |
| *<CLASSNAME>* is present. |
| an H&E stained image of *<CLASSNAME>*. |
| an H&E stained image showing *<CLASSNAME>*. |
| an H&E image showing *<CLASSNAME>*. |
| an H&E image of *<CLASSNAME>*. |
| *<CLASSNAME>*, H&E stain. |
| *<CLASSNAME>*, H&E |

Table 10: Concepts for TCGA-BRCA.

| Concept Type | Classes | Concept Prompts |
|---|---|---|
| Class-specific Concept | IDC | invasive ductal carcinoma
breast invasive ductal carcinoma
invasive ductal carcinoma of the breast
invasive carcinoma of the breast, ductal pattern
idc |
| | ILC | invasive lobular carcinoma
breast invasive lobular carcinoma
invasive lobular carcinoma of the breast
invasive carcinoma of the breast, lobular pattern
ilc |
| Class-agnostic Concept | Adipocytes | adipocytes
adipose tissue
fat cells
fat tissue
fat |
| | Connective tissue | connective tissue
stroma
fibrous tissue
collagen |
| | Necrotic Tissue | necrotic tissue
necrosis |
| | Normal Breast Tissue Cells | normal breast tissue
normal breast cells
normal breast |

Table 11: Concepts for TCGA-NSCLC.

| Concept Type | Classes | Concept Prompts |
|---|---|---|
| Class-specific Concept | LUAD | adenocarcinoma
lung adenocarcinoma
adenocarcinoma of the lung
luad |
| | LUSC | squamous cell carcinoma
lung squamous cell carcinoma
squamous cell carcinoma of the lung
lusc |
| Class-agnostic Concept | Connective tissue | connective tissue
stroma
fibrous tissue
collagen |
| | Necrotic Tissue | necrotic tissue
necrosis |
| | Normal Lung Tissue Cells | normal lung tissue
normal lung cells
normal lung |

Table 12: Concepts for TCGA-RCC.

| Concept Type | Classes | Concept Prompts |
|---|---|---|
| Class-specific Concept | CCRCC | clear cell renal cell carcinoma
renal cell carcinoma, clear cell type
renal cell carcinoma of the clear cell type
clear cell rcc |
| | PRCC | papillary renal cell carcinoma
renal cell carcinoma, papillary type
renal cell carcinoma of the papillary type
papillary rcc |
| | CHRCC | chromophobe renal cell carcinoma
renal cell carcinoma, chromophobe type
renal cell carcinoma of the chromophobe type
chromophobe rcc |
| Class-agnostic Concept | Adipocytes | adipocytes
adipose tissue
fat cells
fat tissue
fat |
| | Connective tissue | connective tissue
stroma
fibrous tissue
collagen |
| | Necrotic Tissue | necrotic tissue
necrosis |
| | Normal Kidney Tissue Cells | normal kidney tissue
normal kidney cells
normal kidney |

## H  More Visualization

To further illustrate the effectiveness of the proposed CATE-MIL, we provide more visualization results in this section. The visualization results for IDC and ILC in TCGA-BRCA are shown in Figure 7 and Figure 8, respectively.

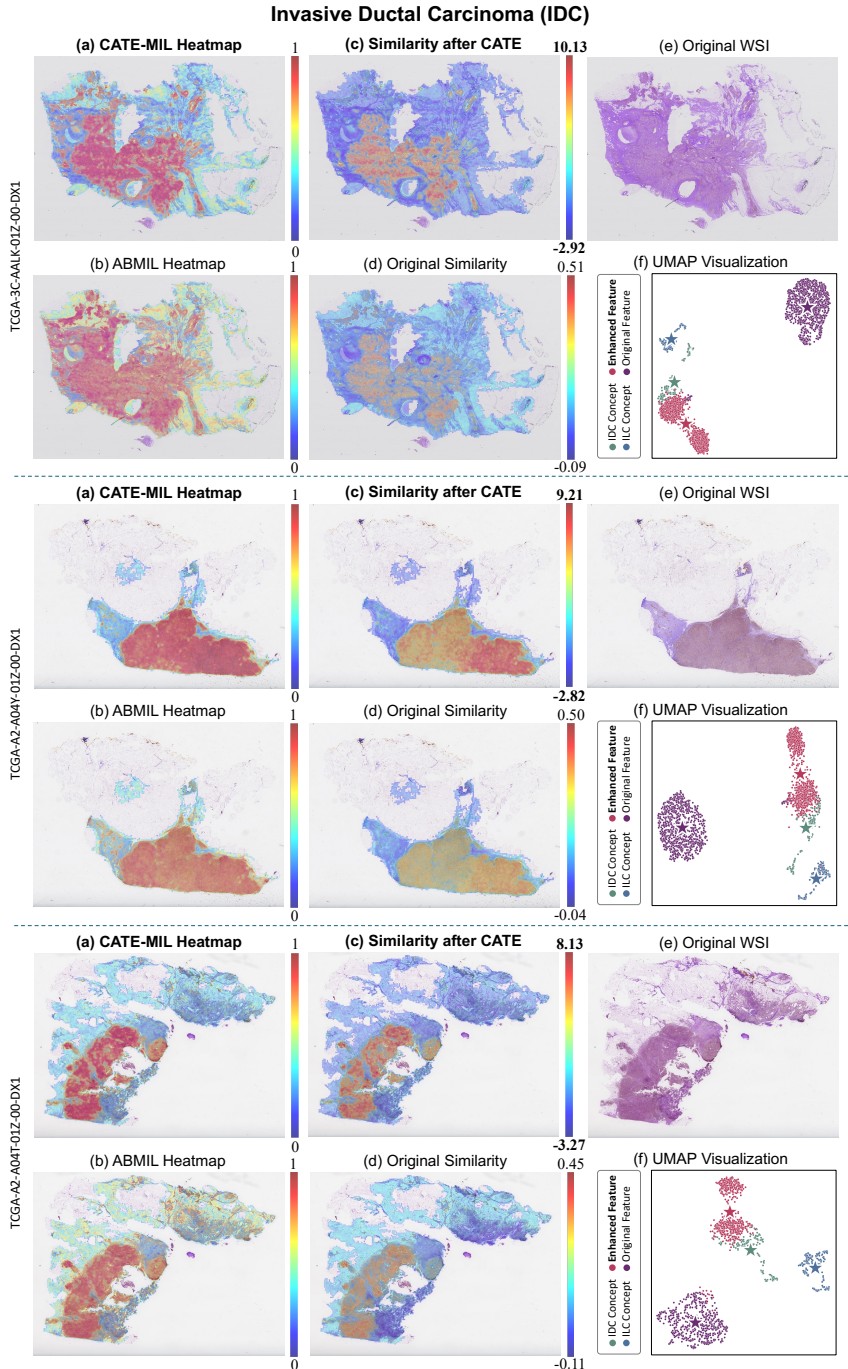

Figure 7: Visualization results for samples of IDC in TCGA-BRCA. (a) Attention heatmap of CATE-MIL. (b) Attention heatmap of original ABMIL. (c) similarity between the calibrated features and the corresponding class concept feature. (d) similarity between the original features and the corresponding class concept feature. (e) Original WSI. (f) UMAP visualization.

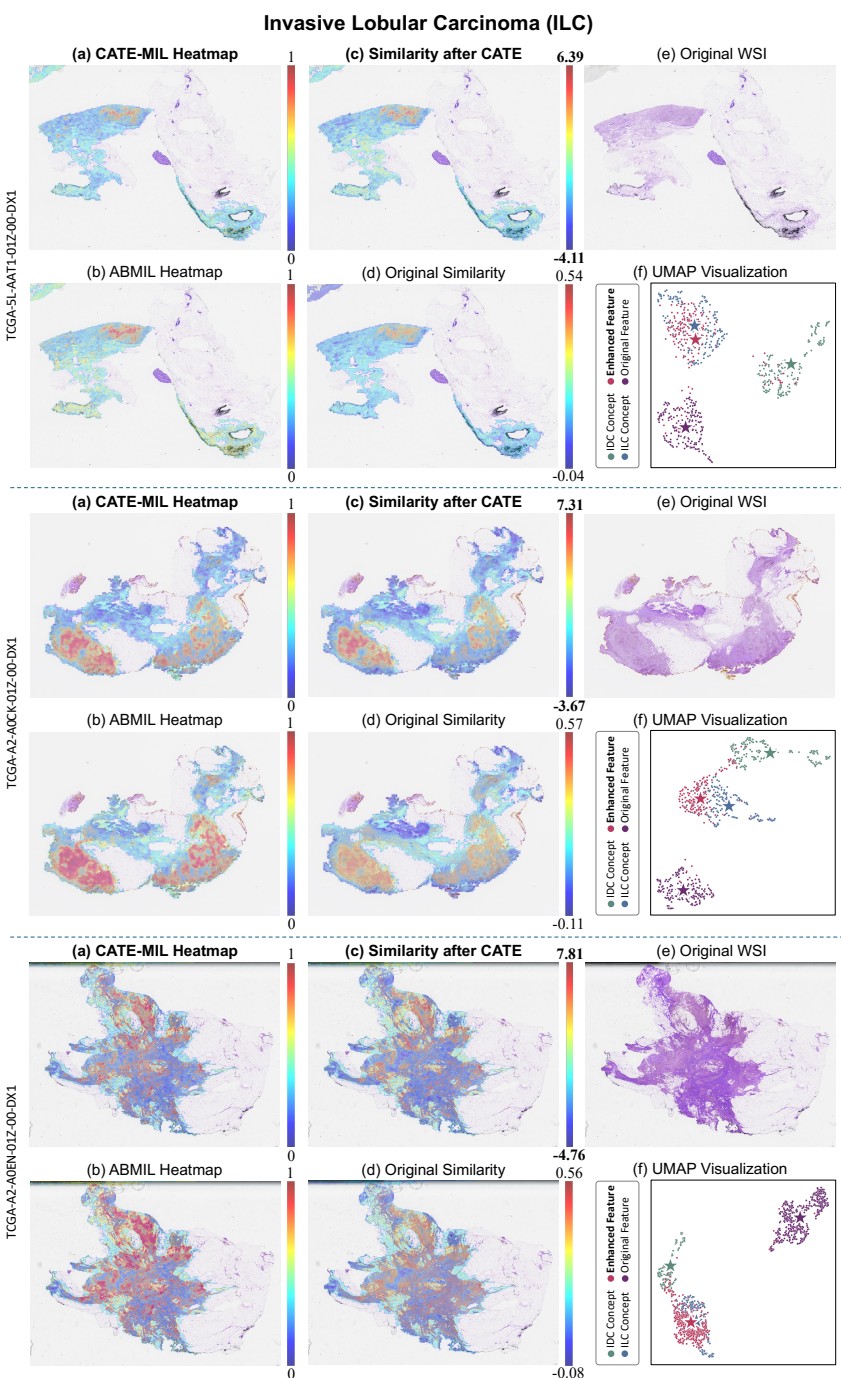

Figure 8: Visualization results for samples of ILC in TCGA-BRCA. (a) Attention heatmap of CATE-MIL. (b) Attention heatmap of original ABMIL. (c) similarity between the calibrated features and the corresponding class concept feature. (d) similarity between the original features and the corresponding class concept feature. (e) Original WSI. (f) UMAP visualization.

