# OpenReview forum: "Free Lunch in Pathology Foundation Model: Task-specific Model Adaptation with Concept-Guided Feature Enhancement"
_NeurIPS.cc/2024/Conference — NeurIPS 2024 poster_

### Official Review · Reviewer_VcAb · 2024-07-08

**Soundness:** 3
**Presentation:** 4
**Contribution:** 2
**Rating:** 5
**Confidence:** 5

**Summary:**

The paper proposes a way to enhance image features from VLM pathology foundation models for downstream tasks by aligning them with task-specific text prompts. The authors use two modules -- one which extracts an image representation aligned with task concepts and other which measures similarity of representation with tasks, to enrich image representations for downstream tasks.
The authors show results for the cancer subtyping task across 3 datasets indicating the usefulness of the proposed approach.

**Strengths:**

**Clarity and Quality**:

The paper is well written and the key ideas are explained clearly.
The experiments are well setup, the results and ablations provide good evidence of the proposed method for the specific problems and the limitations are discussed.
The code is shared and the supplementary section provides more details around ablations and the theoretical assumptions.

**Originality**
There has been prior work in using task anchors to improve MIL using learnt or clustered task prototypes (ProtoMIL, TPMIL, etc)
The idea of using textual prompts to create task anchors and use these to filter patches and enrich patch representations is interesting.
The predictive information maximization (PIM) objective is interesting and is similar to applying CLIP on-the-fly with fixed text embeddings, to extract more task-specific image features.
The SIM objective seems useful to suppress task-irrelevant information.

**Significance**
The proposed approach is a simple addition to MIL and can be applied in conjunction with existing MIL methods.

**Weaknesses:**

The main weaknesses are around some of the assumptions made in the paper listed below which limit the applicability of the approach.


**Aligning patches with tasks**

In WSIs the labels are only available at slide-level and not patch-level and for many problems only a fraction of the patches in the slide have task-relevant info and for many others the information may need to be aggregated from multiple patches.

Even though there is a filtering step to filter patches, the step of aligning patches to tasks is noisy when many patches dont have task info and it may not even be appropriate for problems which need aggregation of information across patches (as the approach homogenizes representations of all patches belonging to the same class).

The authors show results on only one type of task -- subtyping where most patches on the slide have the sub-type information and is also typically a simpler MIL task indicated by the strong results on the benchmarks for most methods.

Instead of aligning patch representations, it would be better to align slide-level representations with task-concepts, which addresses both the drawbacks listed above. There is also evidence from prior works around usefulness of this like SC-MIL (https://arxiv.org/abs/2303.13405 which align representations of slides belonging to same-class) and they also show aligning patch-level representations (patch SC-MIL) does poorly compared to aligning at the slide-level


**Unclear Improvements**

It is unclear how useful the task-alignment objective with the information bottleneck layer is.
The key part seems like the filtering step which extracts relevant patches for the task and I suspect if thats contributing to majority of the improvements.
Once filtering is done, is the alignment step really needed as it seems like a dimensionality reduction step to extract task specific features.
A sufficiently strong model should be able to able to extract task relevant features from the embedding dimensions with cross-entropy loss.
The alignment step might also be unsuitable as it homogenizes patch representations which may not be good for many MIL tasks and directly optimizing with cross entropy on task-labels might be more flexible and general.

Its also unclear if the improvements are coming from a bigger model due to the CIB and CFI modules. Scaling ABMIL to similar number of params would provide more context on this.


**Task Concept Alignment**
The other key assumption is using text prompt embeddings as task anchors which is very dependent on factors like the quality of the pre-trained VLM, the textual prompts and descriptions used and the idea that it is possible to write morphological descriptors for tasks which is typically not possible for many MIL tasks like survival or molecular signature prediction.

The sufficiency and consistency constraints imposed could be limiting specially when any of the factors above break. The SIM objective specifically could be problematic when text doesnt fully capture the required features for the task and other relevant image features could be suppressed.

**Questions:**

Its unclear how much improvements are coming from the patch filtering step and the increased number of parameters and if those alone are sufficient enough.

I think its worth trying the alignment on the slide-level representations instead of patch-level representations (similar to SC-MIL) which makes the approach more generic and limits the drawbacks with using slide-level labels for patches.

**Limitations:**

Highlighted the limitations in weaknesses.

Provided some suggestions and improvements in the questions.

Given the weaknesses and the lack of evidence and limited applicability of the approach to more complex MIL problems like scoring (gleason grading, HER2 grading) which need aggregation of information across patches or other analytical tasks like survival prediction, molecular signature prediction where there might not be possible to write text descriptions, I am basing my decision

---

> ### Author Rebuttal · Authors · 2024-08-07
>
> We sincerely appreciate your detailed and constructive comments. We will address your concerns as follows:
>
> > **1. Aligning patches with tasks**
>
> Thanks for your comments. We address your concerns from the following aspects.
>
> - The motivation behind patch-level alignment in this work is to enhance the patch features extracted by the pathology VLM to adapt them to specific downstream tasks and make the CATE more general and able to be incorporated with any MIL method.
> The alignment is based on the information bottleneck principle and does not explicitly rely on the patch labels. For patches that don't contain task-relevant information, the CIB module will filter out the task-agnostic information, making the alignment more accurate and less noisy.
>
> - We agree that slide-level alignment is more straightforward. However, aligning only at the slide level may be difficult, as the slide-level representation (aggregated from patch features) is noisy without patch-level feature enhancement, hindering the model’s training, as shown in the following table.
> Incorporating the slide-level alignment with our patch-level alignment can further improve the model’s performance, which demonstrates the importance and rationale of patch-level alignment in the CATE.
>
> |Method|Patch-level Alignment|Slide-level Alignment|BRCA($N_{IND}$=1)|BRCA($N_{IND}$=2)|NSCLC($N_{IND}$=2)|NSCLC($N_{IND}$=4)|RCC($N_{IND}$=3)|RCC($N_{IND}$=6)|
> |---|---|---|---|---|---|---|---|---|
> |ABMIL|||0.914±0.015|0.899±0.035|0.874±0.021|0.951±0.023|0.973±0.005|0.971±0.007|
> |Aligning Patch (CATE-MIL w/o CFI)|&#10004;||0.936±0.010|0.942±0.010|0.910±0.030|0.960±0.011|0.979±0.004|0.977±0.005|
> |Aligning Slide||&#10004;|0.899±0.019|0.883±0.025|0.853±0.018|0.945±0.017|0.970±0.004|0.961±0.011|
> |Aligning Patch+Slide|&#10004;|&#10004;|0.941±0.007|0.944±0.011|0.923±0.019|0.969±0.004|0.982±0.002|0.978±0.003|
>
>
> > **2. Unclear improvements**
>
> Thanks for your comments.
> The filtering step alone, without concept alignment, does not provide any benefit, and the increased number of parameters in the CIB and CFI modules does not directly lead to performance improvement.
>
> - The filtering step is used to obtain a subset of the most task-relevant patch features to align with concepts and supervise the training of the information bottleneck. During the training of MIL and the prediction process, all patch features are input to the CIB module to filter task-agnostic information. Without concept alignment, the CIB module degenerates into an encoder of VAE and does not provide any benefit.
>
> - To show this, we conducted experiments on CATE-MIL without concept alignment (discarding PIM loss and SIM loss of the CIB module) and replaced the CIB module with an MLP to investigate the effect of **concept alignment** and the **increased number of parameters**. The results are shown in the following table. The performance of CATE-MIL significantly decreases in both cases, demonstrating the importance of concept alignment in the CIB module and that the improvements of CATE are not due to the increased number of parameters. We will add this results in the revised manuscript.
>
> |Method|BRCA($N_\text{IND}$=1)|BRCA($N_\text{IND}$=2)|NSCLC($N_\text{IND}$=2)|NSCLC($N_\text{IND}$=4)|RCC($N_\text{IND}$=3)|RCC($N_\text{IND}$=6)|
> |---|---|---|---|---|---|---|
> |ABMIL|0.914±0.015|0.899±0.035|0.874±0.021|0.951±0.023|0.973±0.005|0.971±0.007|
> |CATE-MIL|0.936±0.010|0.942±0.010|0.910±0.030|0.960±0.011|0.979±0.004|0.977±0.005|
> |CATE-MIL w/o concept alignment|0.900±0.017|0.884±0.033|0.742±0.059|0.897±0.022|0.961±0.011|0.932±0.016|
> |Replace CIB with MLP|0.888±0.027|0.902±0.037|0.816±0.040|0.931±0.023|0.966±0.006|0.951±0.021|
>
>
> > **3. Task concept alignment and applicability to more complex MIL problems**
>
> - As we pointed out in the original paper, we agree that the performance of CATE relies on the quality of the pathology VLM and it may not be difficult to be applied to other analytical tasks that are difficult to describe with prompts.
> However, we argue that the main contribution of this work is that it offers a novel and promising idea to adapt the pathology VLM to downstream tasks by leveraging the inherent consistency between image and text in pathology VLM.
>
> - The proposed CATE can benefit more complex tasks beyond subtyping, such as Gleason grading. Experiments on the TCGA-PRAD dataset show that CATE enhances Gleason grading performance, demonstrating that CATE does not affect the aggregation of patch features in the MIL model.
>
>
> |Method|OOD-AUC|OOD-ACC|IND-AUC|IND-ACC|
> |---|---|---|---|---|
> |ABMIL|0.704±0.034|0.510±0.075|0.742±0.060|0.575±0.051|
> |CATE-MIL|0.755±0.050|0.567±0.067|0.797±0.044|0.643±0.075|
>
>
> - In the future, as more studies reveal the connection between morphological features and molecular biomarkers and more powerful pathology VLMs are developed, our framework has the potential to benefit more complex tasks.

---

> > ### Comment · Reviewer_VcAb · 2024-08-08
> >
> > > Aligning patches with tasks
> >
> > Thanks to the authors for exploring the suggestion of aligning with slides. It makes sense that without patch-level alignment, the slide-level alignment can be noisy and has a harder task. Good to see the slide-level alignment helping and complementing the patch-level alignment.
> >
> >
> > > Unclear improvements
> >
> > Thanks for runnning the ablations. In CATE-MIL w/o concept alignment, do you filter patches using similarity and then directly use the patch features from the image encoder for ABMIL?
> > In the ABMIL baseline I assume we do not have the patch-filtering step but everything should be similar to the CATE-MIL w/o concept alignment setup right? Any ideas why AB-MIL is doing better than CATE-MIL w/o concept alignment then?
> >
> > In Replace CIB with MLP, do you project the image features to a bottleneck dimension using an MLP and then use these features for ABMIL? Agree without concept alignment the features may not always capture relevent information but surprised to see it doing so poorly compared to ABMIL. Any hypothesis as to why this is the case.
> >
> > > Task concept alignment and applicability to more complex MIL problems
> >
> > Thanks for running the experiments for gleason grading. This is helpful.
> >
> > > Summary
> >
> > Thanks to the authors for running these additional experiments and strengthening the paper. Could you confirm the setup for the ablations discussed above?

---

> > > ### Author Response · Authors · 2024-08-08
> > >
> > > Dear Reviewer VcAb,
> > >
> > > Thank you very much for your thorough review and for taking the time to read our rebuttal carefully. We appreciate your insightful comments and the opportunity to address them further. We would like to address your additional comments below:
> > >
> > > > In CATE-MIL w/o concept alignment, do you filter patches using similarity and then directly use the patch features from the image encoder for ABMIL? In the ABMIL baseline I assume we do not have the patch-filtering step but everything should be similar to the CATE-MIL w/o concept alignment setup right? Any ideas why AB-MIL is doing better than CATE-MIL w/o concept alignment then?
> > >
> > > - To ease your understanding, we will further clarify the key idea of our concept-guided Information Bottleneck (CIB) module.
> > > In CATE-MIL, the CIB module functions as an encoder (similar to the encoder of VAE) to acquire the enhanced features of **all** the original patch features, where the module parameters are optimized via the concept alignment objectives (i.e., PIM loss and SIM loss).
> > > As not all patches contain task-relevant information, we filter out a subset of patches that contain task-specific information and only calculate the objective losses on these selected patches to avoid instability during the optimization.
> > > However, in the prediction process, we take **all** enhanced patch features into ABMIL to get the final slide-level prediction. Therefore, there is no filtering step in the testing phase.
> > > In summary, the "patch filtering-like" step only **occurs in the training phase, not the testing phase**.
> > >
> > > - In CATE-MIL w/o concept alignment, we did not remove the CIB module directly, but only removed the concept alignment objectives (i.e., PIM loss and SIM loss) and used the cross-entropy loss on task labels to optimize the encoder parameters.
> > > **All the original patch features** are still input to the CIB module and then to ABMIL, **without the filtering-like step**. Therefore, the process of CATE-MIL w/o concept alignment resembles ABMIL with an encoder of VAE appended.
> > >
> > > - The results presented in the rebuttal indicate that such a setting does not enhance the performance of ABMIL; instead, it diminishes performance. This could be attributed to the increased difficulty in training ABMIL and the heightened risk of overfitting caused by the additional layers added before ABMIL. Consequently, the performance of CATE-MIL w/o concept alignment is lower than that of ABMIL, which also validates the effectiveness of the proposed concept alignment objectives.
> > >
> > > > In Replace CIB with MLP, do you project the image features to a bottleneck dimension using an MLP and then use these features for ABMIL? Agree without concept alignment the features may not always capture relevent information but surprised to see it doing so poorly compared to ABMIL. Any hypothesis as to why this is the case.
> > >
> > > - Yes, we project the image features to a bottleneck dimension that matches the enhanced feature dimension in the CIB module using an MLP and then use these features for ABMIL.
> > > As we discussed above, the CIB module can be regarded as a encoder of VAE with concept alignment objectives regularizing the training of the encoder. Without concept alignment objectives, the CIB module functions merely as an encoder, similar to an MLP.
> > > In this case, **directly adding such an encoder or MLP before ABMIL without proper "regularization" will increase the difficulty of the training of ABMIL and potentially lead to overfitting.** Therefore, the performance of CATE-MIL w/o concept alignment is similar to that of replacing the CIB module with an MLP, and both underperform compared to ABMIL.
> > >
> > > > Summary
> > >
> > > The "patch filtering-like" step is only used for calculating alignment losses during training and does not affect the prediction process during inference. The performance improvement of our CATE-MIL model comes from the task/concept alignment objectives, not from the patch filtering step. Additionally, the improvement is not due to an increased number of parameters, as shown by the reduced results when replacing the CIB module with a vanilla MLP having a similar number of parameters.
> > >
> > > We hope this clarifies your concerns. If you have any further questions or need additional clarification, please feel free to let us know.

---

> ### Comment · Reviewer_VcAb · 2024-08-11
>
> Thanks to the authors for clarifying the patch-filtering step and the ablations.
> Agree increased complexity can lead to overfitting for ABMIL, although it isnt super clear if CATE-MIL w/o concept alignment has more params overall as the classifiers and attention layers in ABMIL now operate on bottleneck dimension which is lower than original encoder dimension. The results also arent super consistent as replacing CIB with MLP helps performance in some cases.
>
> I appreciate the authors running these additional experiments and ablations. I think its worth adding these to the paper.
>
> I still feel the the step of aligning patches to tasks is noisy and depends a lot of the quality of the VLM, the nature of the tasks (tasks where there is clear morphological feature descriptions), these being represented in the VLM pre-training and the quality of prompts used for generating task/concept anchors.
>
> However I think this provides a useful way to incorporate domain specific knowledge into WSI classification problems when a well trained pathology VLM is available. Based on this I am changing my decision from 4 to 5.

---

> > ### Author Response · Authors · 2024-08-12
> >
> > Dear Reviewer VcAb,
> >
> > Thank you very much for your thoughtful and constructive feedback. We greatly appreciate your positive comments, and your insights have significantly contributed to strengthening our work. We would like to address your remaining minor concerns regarding the feature dimension of the output from the CIB module.
> >
> > The encoder of CIB module does not change the feature dimension of the input patch features. Since the patch features and concept anchors are generated by the image encoder and text encoder of the pathology VLM respectively, they share the same feature dimension.
> > Therefore, the number of parameters in the classifiers and attention layers in CATE-MIL is identical to that in vanilla ABMIL.
> >
> > We hope this clarifies your concerns. And we will add these clarifications and the results of the additional experiments to the revised manuscript or the appendix.

---

> > > ### Comment · Reviewer_VcAb · 2024-08-13
> > >
> > > Thanks for clarifying the question about the feature dimension.
> > > It makes sense that the dimension is kept same to match text embedding, however the usage of "bottleneck" implicated a smaller dimension, worth mentioning this distinction in the paper.

---

> > > > ### Author Response · Authors · 2024-08-13
> > > >
> > > > We apologize for the confusion caused by the term "bottleneck". We will clarify this in the revised manuscript to avoid any misunderstanding. Thank you for pointing this out.

---

### Official Review · Reviewer_R2V2 · 2024-07-10

**Soundness:** 4
**Presentation:** 4
**Contribution:** 4
**Rating:** 8
**Confidence:** 4

**Summary:**

The authors introduce CATE, a novel approach designed to enhance the generalizability of histopathology models by leveraging task-specific concepts derived from the text endoder of pathology vision language models(VLM). CATE includes two modules: CIB and CIF, which work synergistically to improve model robustness. The CIB module identifies and retains features that contribute positively to predicting the task outcome, while it eliminates features that contain irrelevant or superfluous information.  Additionally, CIF specializes in creating task-specific features by leveraging similarities between the enhanced image features the and predefined concept anchors. The authors demonstrate the effectiveness of CATE across an extensive series of histopathology datasets, showcasing superior results in model performance and generalization.

**Strengths:**

- The concept introduced by CATE is highly relevant as it addresses a major challenge in histopathology: ensuring models generalize effectively across diverse datasets and clinical scenarios. In computational pathology, the lack of external validation cohorts often leads to uncertainty about whether models have learned genuine signals or have simply overfitted to confounding factors or batch effects. To validate their approach, the authors split existing datasets into indoor and outdoor test sets, demonstrating CATE's robustness across different conditions.
- The paper is well-presented and clearly written.
- The strength of the paper lies in its innovative approach of repurposing general-purpose vision language models to extract rich, meaningful representations for downsream tasks.

**Weaknesses:**

- CATE's performance hinges largely on the quality of its concept anchors, which in turn depends on domain expertise and the quality of the pre-trained pathology VLM, a weakness that is also highlighted by the authors.

**Questions:**

- Why did you use accuracy as an evaluation metric? Accuracy is straightforward to interpret, but it can be misleading in imbalanced datasets. Metrics like F1-score, weighted accuracy, precision, and recall offer more nuanced insights into model performance, particularly in contexts where class distribution is skewed such as the BRCA dataset.

- What strategy was used to split the dataset into in-domain and out-of-domain test sets? Understanding this process ensures transparency in evaluating the model's generalizability across different clinical settings or data sources.

- How do you define the class-agnostic concepts?

- What is the intuition behind maziming the distance between the enhanced featrures and the class-agnostic concepts? When features are overly tailored to the training samples by maximizing their separation from class-agnostic concepts, the model may become too specialized and prone to overfitting.

**Limitations:**

- Exploring the potential of concept anchors for applications beyond subtyping, such as mutation predictio could be an intersting avenue to explore.

---

> ### Author Rebuttal · Authors · 2024-08-07
>
> We deeply appreciate your positive comments and valuable suggestions. We would like to address your concerns below one by one:
>
> > **1. The quality of concept anchors**
>
> Thank you very much for your understanding. As we discuss in the paper, we agree that CATE's performance highly relies on the quality of concept anchors. Fortunately, some operations can be done to improve the quality of the prompts and concepts.
>
> For example, to ensure the quality of the concepts, we can ensemble multiple prompts with various templates and use their average embedding for each class to generate robust and stable class concepts, reducing the risk that the concepts cannot fully capture the characteristics of the class. Additionally, the quality of the concepts will be further improved with the development of pathology research and pathology VLMs.
>
> > **2. Evaluation metrics**
>
> Thank you very much for your valuable comments. In the original paper, we follow the experiment settings of previous papers [1,2] and chose AUC as the primary metric and accuracy as the secondary metric.
> We agree that the accuracy metric can be misleading in imbalanced datasets. Therefore, we mainly focused on the AUC metric in the original paper. To further improve the reliability of the results, we will supplement the F1-score metric in the revised manuscript.
>
>
> > **3. Dataset split strategy**
>
> Thank you for your suggestion. We would like to provide more relevant details. We define the in-domain (IND) and out-of-domain (OOD) data based on the source sites of the TCGA dataset.
> Specifically, each dataset in the TCGA project contains samples from different source sites. The different source sites have different staining and imaging characteristics, causing feature domain shifts among different sites. Therefore, MIL models trained on one site may not generalize well to others. To better evaluate the model performance, we report the testing performance on IND data (in-domain, the testing and training data are from the same sites) and OOD data (out-of-domain, the testing and training data are from different sites).
>
> For the BRCA dataset, we randomly selected one or two sites with samples from two categories as IND data and used the remaining sites as OOD data.
> For NSCLC (2 categories) and RCC (3 categories), **each site contains samples from only one category**. We randomly selected one or two corresponding sites for **each category** as IND data and used the other sites as OOD data, resulting in 1 or 2 IND sites for BRCA, 2 or 4 for NSCLC, and 3 or 6 for RCC.
> For the IND data, we also randomly split it into training, validation, and testing sets; training the models on the IND training set, and evaluating them on both IND testing sets (IND performance) and OOD data (OOD performance).
>
>
> > **4. Definition of class-agnostic concepts**
>
> In our paper, class-agnostic concepts refer to the common tissues that are agnostic to the classification task. For example, in the cancer subtyping task, different subtypes have subtype-specific attributes directly related to the task, known as class-specific concepts.
> Meanwhile, there are many pieces of information in the WSI that are agnostic to different subtypes, such as adipose tissue, connective tissue, and normal tissues. We define these common tissues as class-agnostic concepts.
>
> > **5. The intuition behind maximizing the distance**
>
> Thanks for your comments. We understand your concern. However, the model will not become too specialized and prone to overfitting the training data. The class-specific and class-agnostic concepts are **general across different domains and are not specific to the training data**.
> By bringing the enhanced features closer to the class-specific concepts and further away from the class-agnostic concepts, the model can avoid distractions unrelated to the classification task and focus on the general task-relevant concepts.
> This prevents overfitting on the training dataset, and the maximization of the distance between enhanced features and class-agnostic concepts helps the model generalize better to out-of-domain data.
>
> > **6. Potential applications beyond subtyping**
>
> Thank you for your valuable suggestions. We have conducted experiments to show that CATE can also benefit to other more complex tasks beyond subtyping like Gleason grading, as shown in the following table.
>
> Other tasks like mutation prediction are indeed more challenging compared to subtyping.
> In the future, as more studies reveal the connection between morphological features and molecular biomarkers or patient prognosis, we will continue to explore whether our CATE can be applied to a wider range of pathology analysis tasks.
>
> |Method|OOD-AUC|OOD-ACC|IND-AUC|IND-ACC|
> |---|---|---|---|---|
> |ABMIL|0.704±0.034|0.510±0.075|0.742±0.060|0.575±0.051|
> |CATE-MIL|0.755±0.050|0.567±0.067|0.797±0.044|0.643±0.075|
>
>
> [1] Transmil: Transformer based correlated multiple instance learning for whole slide image classification. NeurIPS 2021.
>
> [2] Scaling vision transformers to gigapixel images via hierarchical self-supervised learning. CVPR 2022.

---

> > ### Comment · Reviewer_R2V2 · 2024-08-13
> >
> > I appreciate the authors running  additional experiments and ablations and providing clarifications to my concerns. I will keep my rating as it is already high, as I find this work to be both novel and promising in advancing the field of computational pathology.

---

### Official Review · Reviewer_9fs8 · 2024-07-12

**Soundness:** 3
**Presentation:** 3
**Contribution:** 3
**Rating:** 6
**Confidence:** 4

**Summary:**

The authors introduce a new tool called Concept Anchor-guided Task-specific Feature Enhancement (CATE) for analysis of whole slide images.  The authors are taking advantage of the open source weight now available for the CONCH pathology vision language model trained on paired captions of images from journals and other publicly available image-caption pairs.  The high level conceptulization is that language prompts related to a task at the whole slide image level can be enrich for the features most relevant to the downstream task.  The title cleverly asserts this is a free lunch when using a vision language encoder for feature extraction the expertly curated concepts can "bring foward" the most relevant features when aggregating at the whole slide level.

Interestingly, they split up the CATE task into Predictive Information Maximization to enhance features associated with related concepts and Superfluous Information Minimization is used to supress the concepts likely to be associated with unimportant features.

The outputs of PIM and SIM are used during aggregation task to determine weighting that MIL should use for a given feature vector derived from the tile.

Finally a module called concept-feature interference is used calibrate the CIB scores.

These module provide enhanced features to the aggregation function that provides relative weighting based on the relationship between the semantic concepts relationship to the downstream task.

CATE and weights are not backpropogated through the feature embedding tasks and they are static in the MIL function (to the best of my understanding).

The authors chose RCC, NSCLC, and BRCA tasks from the TCGA to demonstrate performance

**Strengths:**

The model is build such that one can introduce to any MIL-like framework that uses a VLM encoder.

It is a clever idea to guide the aggregation function with the concepts one can get from VLM encoder.

The paper is clearly written and claims are not over stated.

On the chosen tasks, it is reasonably clear that a performance boost is acheived consistently and in some cases by a dramatic margin.  Given that is is a bolt-on method these are very good apples to apples comparisons.

The ablations study suggests that the 3 novel modules (PIM, SIM, and CFI and  parts are additive to peformance, at least on these tasks.

**Weaknesses:**

I agree with the weaknesses the authors have put forth.
The model scope is ~mostly~ limited to downstream task for the the concepts are part of the VLM training.  The tasks that are chosen for benchmarking are actually tasks that are likely well represented in VLM training sets with robust descriptions of SCC vs. LUAD, etc.  For tasks that source material does not contain common descriptions, say more challenging tasks like predicting mutations, it is unclear to what extent, this work would provide advantage over other approaches.
My presumption is that VLM encoders are less robust for feature extraction than the now available large foundation models trained with 10^6 plus whole slide images. If one is not able to use these models and must use an inferior model, then one might argue it is not a "free lunch".

**Questions:**

Out of distribution (OOD) is a term that means many different things to different people.  Will you please spare a paragraph to better explain what you mean by OOD. If you find that OOD is not the best term for what you mean, I think it would be better to use a less loaded term.

Have you considered a hybrid approach where the vision feature vectors can be extracted with a different foundation model but the VLM is use for the concept guidance?

How the normal distribution of SIM in equation (8) align with Bernoulli distribution of ABMIL? Are there alternative distributions that could be used here? (may be Poisson distribution to define random occurance of a non-informative patch?)

Minor comments for improvement in readability:
1) In figure 2a, the image encoder cartoon is larger on the side of image which implies the feature vectors are larger than the original patches.
2) Equation (6) LHS should be I(x; alpha | c)

**Limitations:**

Limitations section is reasonable.  I have provided further feedback in the weaknesses section.

---

> ### Author Rebuttal · Authors · 2024-08-07
>
> We highly appreciate your positive feedback and constructive suggestions. We would like to address your concerns as follows:
>
> > **1. More diversified downstream tasks**
>
> Thank you very much for your valuable comments. As we discussed in the paper,  we agree that the performance of CATE relies on the well-presented concepts generated by pathology VLM. For other tasks such as survival prediction, the concepts may be more complex and hard to write, which may limit the applicability of CATE.
> However, CATE has the potential to benefit more complex tasks beyond subtyping like Gleason grading. We have conducted experiments and shown that CATE can enhance Gleason grading performance. We will add the complete results in the revised manuscript.
> In the future, as more studies reveal the connection between morphological features and molecular biomarkers, our framework has the potential to benefit more complex tasks.
>
> |Method|OOD-AUC|OOD-ACC|IND-AUC|IND-ACC|
> |---|---|---|---|---|
> |ABMIL|0.704±0.034|0.510±0.075|0.742±0.060|0.575±0.051|
> |CATE-MIL|0.755±0.050|0.567±0.067|0.797±0.044|0.643±0.075|
>
> > **2. Dependence on VLM encoders**
>
> Thank you very much for your kind comments, we believe that with the development of pathology research and pathology VLMs, pathology VLMs will have broader prospects, and more powerful VLMs will be developed, reducing the gap between the pure pathology foundation models.
>
>
> > **3. Explanation of OOD**
>
> Thank you for your valuable suggestion. In this work, OOD represents **out-of-domain**. We use the datasets from the TCGA project, which contain samples from different **source sites**. The diverse source sites may have different imaging technologies and staining protocols, and each site can be regarded as a domain, causing feature **domain shifts** between different sites. Therefore, we select several sites for training, referred to as in-domain (IND) data. Consequently, data from other sites with different feature distributions are called out-of-domain (OOD) data. We will add this explanation to the revised manuscript.
>
> > **4. Hybrid approach of different foundation models**
>
> Yes, in our preliminary experiments, we have considered a similar hybrid approach that uses a more powerful vision foundation model to extract image features and a more powerful text encoder to extract text features, followed by a bridge layer to project these images and text features into a shared embedding space.
> This approach could bring more flexibility to the model design and improve applicability to more challenging tasks.
> However, the end-to-end training of such a hybrid model is more complex and requires much more computational resources, which is beyond the scope of this work.
> In the future, we will explore more parameter-efficient fine-tuning strategies and continue to explore this direction.
>
>
> > **5. Distribution alignment in equitation (8)**
>
> We apologize for the misunderstanding. In fact, the primary goal of SIM is to minimize superfluous information, which can be seen as a form of regularization and is task-agnostic. It does not need to align directly with the Bernoulli distribution of ABMIL. Additionally, we greatly appreciate your suggestion; using the Poisson distribution to define the random occurrence of a non-informative patch is indeed a very promising idea. We will explore this in our future work.
>
> > **6. Minor comments**
>
> Thank you very much for your valuable suggestions. We will adjust the figure and correct the equation in the revised manuscript.

---

> > ### Comment · Reviewer_9fs8 · 2024-08-12
> >
> > Thank you for pointing out my mistake on OOD being out-of-domain rather than the more common use of OOD.  I maintain my weak accept characterization.  I think that this work is very novel and interesting and will contribute to furthering the field of computational pathology. I also do not see the comments addressing my concerns to be comprehensive enough to warrant upgrading my initial impression.  The most useful tasks for computational pathology are tasks for which pathologists do not yet have reliable features for predictions, such as survival and biomarker prediction. Gleason grading is yet another task that pathologists are very good at and the features are widely known.

---

> > > ### Author Response · Authors · 2024-08-13
> > >
> > > Dear Reviewer 9fs8,
> > >
> > > Thank you very much for recognizing the novelty and potential of our work. We agree that tasks such as survival prediction and biomarker prediction are indeed more challenging due to the difficulty of defining reliable features, and we will explore these tasks in our future work.
> > >
> > > Additionally, there might be a potential solution to address this challenge. For instance, we could leverage LLMs or retrieval-based LLMs to generate descriptive prompts about the general morphological appearance of WSIs for specific cancer types. By asking targeted questions, we can summarize reliable and general morphological descriptions associated with different survival outcomes or biomarker expressions (e.g., “Displays significant lymphocyte infiltration within both the neoplastic epithelium and the stroma” for EBV-positive tumors) and further verify these prompts with pathologists. These prompts can then be used to generate morphological concepts that serve as concept anchors, guiding the alignment of patch features and enhancing the task-relevant morphological features.
> > >
> > > Thank you for your insightful comments, which have encouraged us to think more deeply about potential future improvements to our method. We will include the above clarification and discussion in the revised manuscript.

---

### Official Review · Reviewer_ZzEi · 2024-07-13

**Soundness:** 2
**Presentation:** 3
**Contribution:** 3
**Rating:** 5
**Confidence:** 4

**Summary:**

The authors propose an approach to enhance the performance of Multiple Instance Learning (MIL) models by incorporating concept prompts within the context of pathological Vision-Language Models (VLMs). Two modules, i.e., the Concept-guided Information Bottleneck (CIB) module and the Concept-Feature Interference (CFI) module, are designed to calibrate and inject similarity characteristics between features and concepts. Experiments are conducted on three Whole Slide Image (WSI) datasets, yielding interesting results. The paper includes qualitative and ablation analyses.

**Strengths:**

Technically Sound Method:
The proposed approach demonstrates a well-founded methodology, leveraging concept prompts within pathological Vision-Language Models for enhancing Multiple Instance Learning models.

Well-Written Paper:
The paper is clearly and coherently written, providing a logical flow and structure that makes the complex ideas accessible and easy to follow.

Well-Illustrated Figures:
The figures are well-designed and effectively illustrate key modules, significantly aiding in the understanding of the methodology.

**Weaknesses:**

Dependence on Text Prompts: The performance heavily relies on the quality of text prompts/concepts. A more principled approach to generating expert-designed prompts would be beneficial. Only the hard-crafted ones make the  "free lunch" not that free.

Experimental Settings: The choice of the number of IND and OOD sites for different datasets is unclear. How to choose the number of IND and OOD sites? It would be helpful to see performance under more traditional settings on NSCLC and RCC. What's more, it is noticed that the gain of CATE decreases with an increasing number of sites, raising questions about the practicality of the main experimental settings in real-world WSI analysis.


Attention Maps: The attention maps are not that convincing to me. Since CATE-MIL provides almost identical attention maps to the baseline ABMIL except for intensity. If ABMIL predicts incorrect attention, CATE-MIL could potentially worsen the situation.


Underperformance on NSCLC and RCC: The authors claim that the underperformance of CATE-MIL on NSCLC and RCC due to the elimination of site-specific patterns is concerning. Clarification on why site-specific information for the IID test is not considered task-relevant is needed, especially as this issue does not appear with BRCA.

**Questions:**

The paper is technically sound, but I am not convinced by the provided experimental results.

1) Could the authors suggest more principled ways to obtain expert-designed prompts? Relying on hand-crafted prompts may not be scalable or generalizable.

2) Please refer to the weakness and clarify the experimental settings. More detailed qualitative and quantitative analyses are needed.

3) Lack of comparison with ref [1], which also uses the Information Bottleneck theory to improve the feature quality of MIL.


[1] Task-specific Fine-tuning via Variational Information Bottleneck for Weakly-supervised Pathology Whole Slide Image Classification. CVPR 2023.

---

> ### Author Rebuttal · Authors · 2024-08-07
>
> We sincerely thank you for your insightful comments and constructive suggestions. We would like to address your concerns below:
>
> > **1. Dependence on text prompts and principled ways to obtain prompts**
>
> In our experiments, in addition to the expert-designed 'class-specific prompts' from the original CONCH article, we also employed GPT-4 to generate 'class-agnostic prompts', asking questions such as 'In addition to tumor tissues, what types of tissue or cells are present in whole slide images of breast cancer?'
> The quality of GPT-4 generated prompts has been demonstrated in several recent studies [2,3].
> In principle, we can use LLM (e.g., GPT-4) to generate reliable expert-designed prompts and further verified by pathologists. This strategy can ensure the scalability and reliability of the prompts.
>
> > **2. Experimental Settings**
>
> - As we discussed in the general response, each dataset in the TCGA project contains samples from different source sites. The different source sites have different staining and imaging characteristics, causing feature domain shifts among different sites. Therefore, MIL models trained on one site may not generalize well to others.
> To better evaluate the model performance, we report the testing performance on IND data (in-domain, the testing and training data are from the same sites) and OOD data (out-of-domain, the testing and training data are from different sites).
>
> - For the BRCA dataset, we randomly selected one or two sites as IND data and used the remaining sites as OOD data.
> **For NSCLC (2 categories) and RCC (3 categories) datasets, each site contains samples from only one subtype**. Therefore, we cannot select only one site as IND data, as it will include one category/subtype in the training data. Instead, we randomly selected one or two corresponding sites for **each category** as IND data and used the other sites as OOD data, resulting in 1 or 2 IND sites for BRCA, 2 or 4 for NSCLC, and 3 or 6 for RCC.
>
>
> > **3. Performance under a more traditional setting and gain of CATE**
>
> - As mentioned above, for NSCLC and RCC, different categories correspond to different sites. In a traditional setting (i.e., training and testing data are from the same sites), MIL models tend to **use site-specific features (e.g., staining) as shortcuts** for high performance, rather than identifying useful class-specific features, making performance less reflective of the models’ actual capability.
> Therefore, we did not report the results of the traditional setting on NSCLC and RCC. If necessary, we can provide these results in the comments.
>
> - The gain of CATE on OOD performance is closely related to the performance of the baseline MIL model. As the performance of MIL models has been very high (over 0.95 OOD-AUC for NSCLC) with more training data, further enhancement becomes harder, reducing the gain from CATE. However, **CATE’s benefit remains positive**. As shown in Table 5 in the Appendix, when all sites are used as IND data for the BRCA dataset (under traditional settings), CATE-MIL still outperforms vanilla ABMIL.
>
> > **4. Attention Maps**
>
> We argue that although the attention maps of CATE-MIL and ABMIL are similar, they are **not identical**. CATE enhances pre-extracted patch features by preserving task-relevant information and eliminating task-agnostic information, which can strengthen the attention scores of cancerous patches and reduce the scores of non-cancerous patches.
> As shown in the more detailed attention maps in the **accompanying PDF**, **some non-cancerous patches mistakenly assigned high attention by ABMIL are corrected by CATE-MIL** (e.g., the upper left area of the second sample), demonstrating CATE-MIL's ability to correct ABMIL's incorrect predictions.
>
> > **5. Underperformance in NSCLC and RCC**
>
> - Site-specific information such as staining style and imaging characteristics is not relevant to the WSI analysis tasks, while it can be used to distinguish different sites.
> For the NSCLC and RCC datasets, as each site contains samples from only one category/subtype, the model can use these site-specific features as shortcuts for subtyping. In other words, the model learns **how to distinguish different sites, not different subtypes**.
> Our CATE-MIL eliminates these site-specific features, so the performance is lower than other MIL models using these spurious site-specific shortcuts. However, our method is more generalizable, as evident by the OOD performance.
>
> - This issue is not very severe in the BRCA dataset, as each site contains samples from two different categories simultaneously. The MIL models need to learn to classify based on task-relevant features rather than site-specific ones. Therefore, CATE's elimination of site-specific patterns does not decrease BRCA's IND performance but improves it by enhancing task-relevant features.
>
> > **6. Comparison with ref [1]**
>
> The comparison results between CATE with ref [1] are shown below. It can be observed that ref [1] underperforms vanilla ABMIL on OOD data for NSCLC and RCC. This shows that ref [1] overfits IND data and focuses on shortcuts rather than task-relevant features. Furthermore, it involves three separate stages, making it time-consuming and resource-intensive.
>
> ||Method|OOD-AUC|
> |---|---|---|
> |BRCA($N_{\text{IND}}$=1)|ABMIL|0.914±0.015|
> ||ABMIL+ref [1]|0.934±0.003|
> ||ABMIL+CATE|0.951±0.003|
> |NSCLC($N_{\text{IND}}$=2)|ABMIL|0.874±0.021|
> ||ABMIL+ref [1]|0.713±0.009|
> ||ABMIL+CATE|0.945±0.016|
> |RCC($N_{\text{IND}}$=3)|ABMIL|0.973±0.005|
> ||ABMIL+ref [1]|0.953±0.004|
> ||ABMIL+CATE|0.983±0.002|
>
> [1] Task-specific fine-tuning via variational information bottleneck for weakly-supervised pathology whole slide image classification. CVPR 2023.
>
> [2] The rise of ai language pathologists: Exploring two-level prompt learning for few-shot weakly-supervised whole slide image classification. NeruIPS 2023.
>
> [3] Generalizable Whole Slide Image Classification with Fine-Grained Visual-Semantic Interaction. CVPR 2024.

---

> > ### Comment · Reviewer_ZzEi · 2024-08-09
> >
> > I appreciate the detailed responses provided. Based on the responses, I still have further questions:
> > 1. From my perspective, using LLMs just transforms the manually designed prompts into questions.  I assume that both of them need a certain level of pathological background, which is totally different from natural images and makes the proposed method not that free.
> > 2. The site-specific signatures have been thoroughly discussed in ref[1]. I recommend using the site-preserved CV for more convincing results instead of 10 runs of Monte-Carlo CV, where higher means and fewer stds are expected for better performance.
> > 3. For the performance, the authors attribute to the class-specific features and the site-specific features. Any more direct evidence to support this argument?
> > 4. Besides this cherry-picked sample, I am still not convinced. I think the the attention maps of CATE-MIL and ABMIL are still almost identical after binarized.
> >
> > [1] The impact of site-specific digital histology  signatures on deep learning model accuracy and  bias, nature communications

---

> > > ### Author Response · Authors · 2024-08-10
> > >
> > > Dear Reviewer ZzEi,
> > >
> > > We appreciate your detailed reviews and valuable comments. We would like to address your additional concerns below:
> > >
> > > > From my perspective, using LLMs just transforms the manually designed prompts into questions...
> > >
> > > Thank you for your feedback. In response to your comments:
> > > - The questions we designed are more general and do not require an extensive pathological background. They are intended to allow the LLM or retrieval-based LLM to utilize embedded domain knowledge in the model or the knowledge in the literature for high-quality prompts.
> > > - We understand that designing such questions is not without effort and we will relax the "free-lunch" claim and discuss it in our revised manuscript.
> > > - We want to emphasize that medical imaging analysis and computational pathology are complex fields requiring significant domain expertise. Incorporating such knowledge/background into the method design could enhance the performance and robustness of the framework. Our approach offers a principled approach to integrating domain knowledge/background into framework design.
> > >
> > > > The site-specific signatures have been thoroughly discussed in ref[1]. I recommend using the site-preserved CV...
> > >
> > > Thank you for your constructive suggestion. We conducted experiments using the suggested site-preserved CV, following the original code and settings from ref [1] to split the data into three folds based on the source sites. The results show that CATE-MIL consistently outperforms vanilla ABMIL across all datasets. We will add these results to the revised paper or appendix (if space is not allowed).
> > >
> > > ||Method|AUC|
> > > |---|---|---|
> > > |BRCA|ABMIL|0.912±0.012|
> > > ||CATE-MIL|0.935±0.014|
> > > |NSCLC|ABMIL|0.953±0.018|
> > > ||CATE-MIL|0.965±0.011|
> > > |RCC|ABMIL|0.980±0.001|
> > > ||CATE-MIL|0.983±0.000|
> > >
> > > > For the performance, the authors attribute to the class-specific features and the site-specific...
> > >
> > > Thanks for your comments. We try to provide more evidence from the following two aspects:
> > > - To evaluate the influence of site-specific features, we selected two sites (site IDs 49 and 55, respectively) with the largest number of samples of **the same subtype (LUAD)** from the NSCLC dataset.
> > > We used the corresponding site ID as labels for the samples and trained an ABMIL model to predict which site the samples originated from. The results show that the ABMIL model can easily predict the site with an AUC of **0.988**. This indicates that the model can easily use site-specific features to distinguish different sites, which may be used as shortcuts for subtyping.
> > > - We also conducted a comparative experiment on the NSCLC dataset using patch features extracted after **stain normalization**, which helps reduce the impact of staining styles and imaging characteristics.
> > > As shown in the following table, the IND performance is reduced. This demonstrates the effect of site-specific features on shortcutting subtyping and hindering the generalization of the model. Meanwhile, the OOD performance of ABMIL is enhanced after stain normalization, indicating that without site-specific features, the model can generalize better to OOD data, which is consistent with CATE-MIL.
> > > Notably, CATE-MIL still outperforms ABMIL in OOD performance after stain normalization, validating that CATE can better eliminate site-specific features to enhance the generalization of the model compared to stain normalization.
> > >
> > > ||Method|OOD-AUC|IND-AUC|
> > > |---|---|---|---|
> > > |NSCLC($N_{\text{IND}}$=2)|ABMIL w/o stain norm|0.874±0.021|0.997±0.004|
> > > ||ABMIL w/ stain norm|0.889±0.019|0.995±0.006|
> > > ||CATE-MIL|0.945±0.016|0.985±0.011|
> > > |NSCLC ($N_{\text{IND}}$=4)|ABMIL w/o stain norm|0.951±0.023|0.974±0.018|
> > > ||ABMIL w/ stain norm|0.959±0.021|0.966±0.026|
> > > ||CATE-MIL|0.969±0.003|0.967±0.019|
> > >
> > > > Besides this cherry-picked sample, I am still not convinced. I think the the attention maps of...
> > >
> > > We apologize for any remaining doubts regarding the attention maps. As discussed in the previous response and supported by the visual evidence in the attached PDF, we have shown that the attention maps of CATE-MIL and ABMIL are not identical.
> > > The attention map of CATE-MIL is more focused on cancerous patches and less on non-cancerous patches. To further clarify this, we calculated the average **Jaccard index** between the binarized attention maps of CATE-MIL and ABMIL for **all samples** to provide the **quantitative evidence**.
> > >
> > > We first binarized the attention maps by setting the threshold to 0.5, and then calculated the Jaccard index by dividing the intersection of the two binarized attention maps by their union. The results show that the average Jaccard index on the BRCA dataset is **0.699**, indicating that the overlap between the attention maps of CATE-MIL and ABMIL is not very high and that the attention maps of CATE-MIL are indeed different from those of ABMIL.
> > >
> > > [1] The impact of site-specific digital histology signatures on deep learning model accuracy and bias, nature communications

---

### Author Rebuttal · Authors · 2024-08-07

We are sincerely grateful to the reviewers for your insightful comments, constructive suggestions, and acknowledging the clarity and quality of our paper, as well as our contributions in terms of novelty and effectiveness in feature enhancement for better WSI analysis.
We have carefully read and considered all the comments and suggestions from the reviewers. We would like to first respond to the most common concerns raised by the reviewers as follows:

> **1. Experimental Settings on IND and OOD Data**:

To help readers better understand our experimental settings, we would like to provide more background and motivation behind the split of in-domain (IND) and out-of-domain (OOD) data for each dataset and the underperformance of CATE on NSCLC and RCC datasets.

- The dataset (e.g., BRCA, NSCLC, and RCC) in the TCGA contains samples from different source sites (i.e., different hospitals or laboratories). **Different source sites have different staining protocols and imaging characteristics**, causing feature domain shifts between different sites [1,2]. Therefore, MIL models trained on one or more sites may not generalize well to others. To better evaluate the true performance of the models, we selected several sites as IND data (in-domain, the testing and training data are from the same sites), and used data from other sites as OOD data (out-of-domain, the testing and training data are from different sites), and reported the testing performance on both IND and OOD data.

- For the BRCA dataset, we randomly selected one or two sites as IND data and used the remaining sites as OOD data. **For NSCLC (2 categories) and RCC (3 categories) datasets, each site contains samples from only one subtype**. Therefore, we cannot select only one site as IND data, as it will include one category/subtype in the training data. Instead, we randomly selected one or two corresponding sites for **each category** as IND data and used the other sites as OOD data, resulting in 1 or 2 IND sites for BRCA, 2 or 4 for NSCLC, and 3 or 6 for RCC.

- Site-specific information such as staining style and imaging characteristics is not relevant to the WSI analysis tasks, while it can be used to distinguish different sites. For the NSCLC and RCC datasets, as each site contains samples from only one category/subtype, the model can use these site-specific features as shortcuts for subtyping. In other words, the model learns **how to distinguish different sites, not different subtypes**. Our CATE-MIL eliminates these site-specific features, so the IND performance is lower than other MIL models using these spurious site-specific shortcuts. However, our method is more generalizable, as evident by the OOD performance.
In contrast, for the BRCA dataset, each IND site contains samples from two different categories simultaneously, requiring MIL models to classify based on task-relevant features rather than site-specific ones. Therefore, CATE's elimination of site-specific patterns does not decrease BRCA's IND performance but improves it by enhancing task-relevant features.

> **2. Application beyond Subtyping**:

As we pointed out in the original paper, CATE is optimized for classification tasks such as cancer subtyping. Moreover, CATE has the potential to benefit more complex tasks beyond subtyping, such as Gleason grading. We have conducted experiments and shown that CATE can enhance Gleason grading performance. We will add the complete results in the revised manuscript. In the future, as more studies reveal the connection between morphological features and molecular biomarkers and more powerful pathology VLMs are developed, our framework has the potential to benefit more complex tasks.
|Method|OOD-AUC|OOD-ACC|IND-AUC|IND-ACC|
|---|---|---|---|---|
|ABMIL|0.704±0.034|0.510±0.075|0.742±0.060|0.575±0.051|
|CATE-MIL|0.755±0.050|0.567±0.067|0.797±0.044|0.643±0.075|

[1] Robust whole slide image analysis for cervical cancer screening using deep learning. Nature communications.

[2] Deep learning-based transformation of H&E stained tissues into special stains. Nature communications.

[3] The rise of ai language pathologists: Exploring two-level prompt learning for few-shot weakly-supervised whole slide image classification. NeruIPS 2023.

[4] Generalizable Whole Slide Image Classification with Fine-Grained Visual-Semantic Interaction. CVPR 2024.

---

### Decision · Program_Chairs · 2024-09-25

**Decision:**

Accept (poster)

**Comment:**

The paper introduces Concept Anchor-guided Task-specific feature Enhancement (CATE), a novel approach that enhances the performance of pathology foundation models by tailoring them to specific downstream tasks. By leveraging task-specific concepts derived from vision-language models and incorporating a Concept-guided Information Bottleneck module, the method effectively maximizes the relevance of extracted features while minimizing irrelevant information. Adding a Concept-Feature Interference module further refined the discriminativeness of these features. The reviewers acknowledged the novelty and potential impact of this approach, especially its ability to incorporate domain-specific knowledge into whole slide image (WSI) classification. While some concerns were raised regarding the dependency on vision-language model quality and potential overfitting, the additional experiments and clarifications provided by the authors in their rebuttal were sufficient to address most of these issues. The consensus is to accept this paper.